

# Above-Cloud Aerosol Radiative Effects based on ORACLES 2016 and ORACLES 2017 Aircraft Experiments

Sabrina P. Cochrane[1,2], K. Sebastian Schmidt[1,2], Hong Chen[1,2], Peter Pilewskie[1,2], Scott Kittelman[1], Jens Redemann[3], Samuel LeBlanc[4], Kristina Pistone[4], Meloë Kacenelenbogen[4],Michal Segal Rozenhaimer[4], Yohei Shinozuka[5], Connor Flynn[6], Steven Platnick[7], Kerry Meyer[7], Rich Ferrare[8], Sharon Burton[8], Chris Hostetler[8], Steven Howell[9], Amie Dobracki[10], Sarah Doherty[11]

[1]Department of Atmospheric and Oceanic Sciences, University of Colorado, Boulder, 80303, USA

[2]Laboratory for Atmospheric and Space Physics, Boulder, 80303, USA

[3]School of Meteorology, University of Oklahoma, Norman, Oklahoma, 73019, USA

[4]Bay Area Environmental Research Institute/NASA Ames Research Center, Mountain View, 94035, USA

[5]Universities Space Research Association/NASA Ames Research Center, Mountain View, 94035, USA

[6]Pacific Northwest National Laboratory, Richland, Washington, 99354, USA

[7]NASA Goddard Space Flight Center, Greenbelt, MD, 20771, USA

[8]NASA Langley Research Center, Hampton, VA, 23666, USA

[9]Departmentof Oceanography, University of Hawaii, Honolulu, HI, 96844, USA

[10]Department of Atmospheric Science, Rosentiel School of Marine and Atmospheric Science, University of Miami, Miami, FL, 33146, USA

[11]Joint Institute for the Study of Atmosphere and Ocean, University of Washington, Seattle, WA, 98195, USA

*Correspondence to*: Sabrina Cochrane (sabrina.cochrane@colorado.edu)

**Abstract.** Determining the direct aerosol radiative effect (DARE) of absorbing aerosols above clouds from satellite observations alone is a challenging task, in part because the radiative signal of the aerosol layer is not easily untangled from that of the clouds below. In this study, we use aircraft measurements from the NASA ObseRvations of CLouds above Aerosols and their intEractionS (ORACLES) project in





the southeast Atlantic to derive it with as few assumptions as possible. This is accomplished by using spectral irradiance measurements (Solar Spectral Flux Radiometer, SSFR) and aerosol optical depth (AOD) retrievals (Spectrometer for Sky-Scanning, Sun-Tracking Atmospheric Research, 4STAR) during vertical profiles (spirals) that minimize the albedo variability of the underlying cloud field – thus

isolating aerosol radiative effects from those of the cloud field below. For two representative cases, we retrieve spectral aerosol single scattering albedo (SSA) and the asymmetry parameter (g) from these profile measurements, and calculate DARE given the albedo range measured by SSFR on horizontal legs above clouds. For mid-visible wavelengths, we find SSA values from 0.80-0.85, and a significant spectral dependence of g. As the cloud albedo increases, the aerosol increasingly warms the column.

The transition from a cooling to a warming top-of-aerosol radiative effect (the critical albedo) occurs just above 0.2 in the mid-visible. In a companion paper, we use the techniques introduced here to generalize our findings to all 2016 and 2017 measurements, and parameterize aerosol radiative effects.

# 1 Introduction

## 1.1 Background

Aerosols are ubiquitous throughout the Earth's atmosphere, and they play a crucial role in modulating the flux of solar radiation that reaches the Earth's surface. The energy distribution within a scene that contains aerosols depends not only on the amount of incoming solar radiation, aerosol optical depth (AOD) and type, but also on the albedo beneath the aerosols. Depending on the type of aerosol, the incoming radiation will be absorbed or scattered in a certain ratio, described by the single scattering

albedo (SSA), while the direction (forward or backward) of the scattered radiation can be approximated by the asymmetry parameter (g). Aerosol absorption and scattering change the radiative balance relative to the aerosol-free atmosphere. This perturbation is called the direct aerosol radiative effect (DARE). The scene albedo below an aerosol layer, whether from clouds, ocean or land, can determine whether the layer has a negative (positive) DARE, resulting in a cooling (warming) effect at the top of the

atmosphere (Twomey, 1977; Russell et al., 2002). Aerosols injected into the global climate system by human activity since the beginning of industrialization may offset up to 50% of the warming due to





anthropogenic greenhouse gas emissions (Myhre et al., 2013). However, the uncertainty of this offset is large, in part due to observational challenges: radiative anthropogenic aerosol forcing could range from -1 W m$^{-2}$ to +0.2 W m$^{-2}$ (table 15 in Myhre et al., 2013).

Deriving the direct effect of aerosols on the radiation budget, ignoring for the moment the impact on

radiative balance due to aerosol influences on cloud properties and lifetime, is difficult since DARE is derived from the difference between radiative fluxes in the presence of and absence of aerosol. It is impossible to observe both states simultaneously, and therefore, DARE is not directly measurable and, in most cases, requires a radiative transfer model (RTM) initialized with observational or model inputs of aerosol AOD, SSA, and g as well as the spectral reflectance or albedo below the aerosol layer. The

DARE calculations are limited by the accuracy of the observations and the model accuracy itself. For conditions where absorbing aerosols overlie inhomogeneous cloud fields, determining DARE is even more challenging since the calculations require both the aerosol properties as well as the cloud properties, primarily the cloud spectral albedo. The cloud radiative signal can be relatively large compared to that of aerosol particles. Therefore, it can be difficult to isolate the aerosol radiative effect

from that of clouds, especially when the cloud albedo varies in the sampling region.

## 1.2 Satellite-Derived Cloud and Aerosol Properties to Derive DARE

Obtaining the necessary cloud and aerosol parameters from satellite instruments provides the flexibility to estimate DARE in nearly any region. Until recently, aerosol and cloud properties could not typically

be measured from the same satellite when the aerosol occurs above the clouds, and the strategy to estimate DARE for these conditions was to combine properties from multiple satellites (e.g., Chand et al., 2010; Meyer et al., 2013; Zhang et al., 2016; Sayer at al. 2016; Kacenelenbogen et al., 2018). The problem with this approach, however, is that biases in the cloud and aerosol properties translate into biases in DARE if left unaccounted for (Meyer et al. 2013). For example, many DARE studies utilize

Moderate Resolution Imaging Spectroradiometer (MODIS) cloud optical thickness (COT) and effective droplet size (translated into cloud albedo, which cannot be directly measured from space) and/or AOD from the active lidar instrument Cloud-Aerosol Lidar with Orthogonal Polarization (CALIOP).



However, MODIS cloud retrievals can be biased when absorbing aerosols are present above cloud (Haywood et al., 2004; Wilcox et al., 2009; Coddington et al., 2010) and CALIOP AOD which was known to be low-biased for daytime measurements (Winker et al., 2012; Meyer et al., 2013; Jethva et al., 2014) until the development of a new method to derive AOD above cloud that uses the cloud returns

to derive a much more accurate measure of AOD above cloud (Kim et al., 2018)

Work has been done to characterize and correct for the biases in cloud and aerosol properties in DARE estimates (Meyer et al., 2013; Zhang et al., 2016). Meyer et al. (2015) accounts for the satellite cloud optical property bias by developing a simultaneous retrieval of cloud optical thickness and effective radius and aerosol AOD from MODIS imagery alone, thus obtaining both aerosol and cloud properties

from a single instrument that are used as inputs into DARE calculations. Jethva et al. (2013) also retrieves AOD and COT from MODIS alone, using the color ratio method to derive DARE.

Table 1 in Kacenelenbogen et al. (2018) provides a summary of DARE studies and the methods used to obtain aerosol and cloud properties, and it is clear that although methods to account for satellite AOD and COT biases have been established, the aerosol SSA and g remain difficult to obtain. Often, SSA

and g are obtained from an assumed aerosol model, such as the MODIS MOD04 absorbing aerosol model used in both Meyer et al. (2013) and Meyer et al. (2015), or the CALIOP aerosol sub-type models used by Zhang et al. (2016). This approach requires the correct aerosol model be chosen and some studies choose instead to use optical properties from an outside source. For example, Chand et al. (2009) combine CALIPSO aerosol AOD and Angstrom exponent with MODIS COT, but assume a

regional mean value of SSA from the Southern African Regional Science Initiative (SAFARI) 2000 campaign to derive diurnal DARE. Jethva et al. (2013) estimates DARE using the SSA obtained from Aerosol Robotic Network (AERONET) sites. Since these measurements of SSA are not taken in conjunction with the other cloud and aerosol properties, it is difficult to determine whether they are valid and consistent for the specific aerosol measured by the satellite.



## 1.3 Estimates of DARE from Aircraft Observations

Aircraft observations, as opposed to satellite remote sensing, provide in situ observations of clouds and aerosols that are better suited for deriving their radiative properties, especially when the clouds are inhomogeneous. For example, an aircraft can fly through an aerosol layer to measure aerosol

absorption, scattering, and SSA with in situ instruments or fly directly above a cloud layer to measure the albedo. Studies such as Pilewskie et al. (2003), Redemann et al. (2006), Schmidt et al. (2010), Coddington et al. (2010), LeBlanc et al. (2012), and Bierwirth et al. (2017) along with the work presented here have capitalized on this versatility and developed new algorithms and instrumentation to determine aerosol and cloud properties, which can then be utilized to estimate DARE.

For example, under the specific conditions of an aerosol layer with a loading gradient above a homogenous, dark surface, Redemann et al. (2006) derived the below-layer aerosol forcing efficiency (radiative effect per mid-visible AOD) from the co-varying irradiance/AOD pairs along a leg with minimal dependence on radiative transfer calculations. This method, however, is not applicable for scenes with absorbing aerosols above clouds such as those encountered during the recent NASA

ObseRvations of CLouds above Aerosols and their intEractionS (ORACLES) project (Zuidema et al., 2016). The ORACLES project conducted three aircraft campaigns in the Southeast Atlantic, providing measurements in a region with high biomass burning aerosol loading where there has been few extensive field observations to date. In this study, we combine data from multiple instruments to retrieve the aerosol and cloud properties as directly as possible in order to calculate DARE and

investigate the relationship between DARE and cloud albedo. The sensitivity of DARE above the aerosol layer to the underlying surface can be described by the transition from negative to positive radiative effect, or cooling to warming (Russell et al., 2002). The albedo where this transition occurs, hereafter called the critical albedo, expands upon the quantities of critical reflectance and critical surface albedo that more specifically refer to the relationship between AOD and top of atmosphere

reflectance (Fraser and Kaufman, 1985; Seidel and Popp, 2012). The dependence of the sign of the aerosol's radiative effect on the underlying albedo has been shown for aerosols above clouds in the Southeast Atlantic by Keil and Haywood (2003), Chand et al. (2009), and Meyer et al. (2013).



ORACLES aircraft observations make up an extensive dataset that can be used to validate current satellite methods of deriving the aerosol and cloud properties that go into calculations of DARE. To begin this process, our primary objective of this paper is to derive DARE as a function of (a) the aerosol optical properties and (b) cloud albedo from the ORACLES measurements. In Section 2, we describe

the observations themselves and the sampling approaches used to obtain them. Section 3 describes the methods used to determine SSA and g, and how we utilize the results to calculate DARE as directly as possible. Section 4 presents our findings, while section 5 provides a discussion and ways in which we will explore DARE's dependence on aerosol properties in the future, along with prospective satellite validation goals.

## 2 Observations: Measurement Techniques, Instrumentation, and Data

### 2.1 ORACLES

The first two deployments of the NASA ORACLES experiment were conducted from Namibia in 2016 and from São Tomé in 2017, regions located on or just off the western coast of the African continent. The Southeast Atlantic Ocean is often covered by a seasonal stratocumulus cloud deck capped by a

thick layer of biomass burning aerosols advected from the interior of the African continent, providing ideal natural conditions to assess aerosol radiative effects above various cloud scenes and improve the understanding of many aspects of cloud-aerosol interactions.

Both the NASA P-3 and the ER-2 aircraft were deployed in the 2016 campaign. The P-3 flew at approximately 5 km altitude and below, carrying a comprehensive payload of both in situ and remote

sensing instruments. The ER-2 flew at high altitude, approximately 20 km, carrying remote sensing instruments such as the enhanced MODIS Airborne Simulator (eMAS) and the High Spectral Resolution Lidar 2 (HSRL-2) that collected simultaneous and collocated measurements with the P-3 during several coordinated flights. During the 2016 deployment, the P-3 completed fourteen science flights in total, five of which were collocated with the ER-2, and nine of which included radiation-

specific sampling maneuvers. Although the ER-2 did not participate in the 2017 deployment, the P-3 payload remained nearly the same except for the addition of HSRL-2 that had been deployed on the ER-





during the 2016 campaign. Therefore, we focus on utilizing measurements taken from the P-3, which conducted 12 science flights in total, with five flights dedicated to radiation-specific studies in 2017. In this study, we primarily use measurements taken by SSFR, 4STAR and HSRL-2 to investigate two cases, 20 September 2016 and 13 August 2017, which met specific requirements such as varying scene
albedos and large aerosol loading. A companion paper will present more generalized results.

## 2.2 SSFR and ALP

The SSFR (Pilewskie et al., 2003; Schmidt and Pilewskie, 2012) is comprised of two pairs of spectrometers. Each pair consists of one spectrometer that is sensitive over the near-ultraviolet, visible
and very near-infrared wavelength range, and another that is sensitive in the shortwave infrared wavelength range. The spectra are joined at 940 nm to provide a full spectral range from 350–2100 nm. The SSFR measures downward spectral irradiance ($F_\lambda^\downarrow$) from a zenith light collector mounted on a stabilizing platform on the upper fuselage and upward spectral irradiance ($F_\lambda^\uparrow$) from a nadir light collector fix-mounted to the aircraft. The externally mounted light collectors are connected by fiber
optic cables to the spectrometer, which resides in the aircraft cabin along with the data acquisition unit. The SSFR was radiometrically calibrated with a NIST-traceable 1000 W lamp light source before and after each deployment, and relative calibration changes throughout the field campaign were monitored with a portable field standard. The light collectors consist of an integrating sphere with a circular aperture on top. They weigh the incoming radiance according to an angular response close to the cosine
of the incidence angle. These light collectors have been improved over time (Kindel, 2010) to minimize the dependence on the azimuth angle of the incident radiance. However, the dependence on the polar angle, termed the cosine response, still requires careful characterization in the laboratory before and after the deployment. After applying all corrections, the uncertainty of the SSFR measurements is 3–5% across the spectral range for both zenith and nadir irradiance. More importantly for this study, the
precision is 0.5-1.0%.

The zenith light collector of SSFR was kept horizontally aligned by counteracting the variable aircraft attitude with an Active Leveling Platform (ALP), which was developed at CU Boulder for the NASA C-130 aircraft (Smith et al., 2017) and later rebuilt for the P-3, specifically for ORACLES. ALP relies on



aircraft attitude information from a dedicated Inertial Navigation System (INS) that monitors the aircraft attitude, specifically the pitch and roll angles. This information is sent to a real-time controller, which additionally has the ability to instead ingest data from the aircraft INS. The controller drives the two actuators of a 2-axis tip-tilt stage: one axis for aircraft roll movements and one for aircraft pitch
movements. As the attitude angle changes, the tip-tilt stage adjusts accordingly to maintain the SSFR at the horizontal level position within approximately 0.2 degrees. The nadir light collector is not actively leveled.

For fix-mounted light collectors, not only will the downward irradiance be referenced to an incorrect zenith due to the polar angle of incident light referenced to the aircraft horizon rather than true-
horizontal, but radiation from the lower hemisphere will also contaminate the zenith irradiance measurements if the receiving plane is not properly aligned with the horizon.  This is especially problematic over bright surfaces such as snow, ice, or clouds. For ORACLES, it was important to sample the dependence of the downwelling irradiance on the aerosol conditions above. Since the aerosol-induced irradiance changes are small compared to the reflection by clouds, even minor
contamination from the lower hemisphere could cause a bias in the signal. Such biases cannot be corrected in post-processing because common correction schemes assume that no radiation originates in the lower hemisphere (Bucholtz et al., 2008). ALP alleviates these problems and enables the collection of irradiance data during spiral measurements as long as pitch and roll stay within the ALP operating range of 6°. For the reasons mentioned above, spiral data have traditionally not been useful for radiation
science. In this study, they turn out to be the key for achieving our stated goals.

**2.3 4STAR, HSRL-2, and eMAS**

The 4STAR instrument provides direct-beam measurements of AOD above the aircraft at hundreds of wavelengths ranging from 350 nm to 1650 nm, with a subset of 24 wavelengths available in the main
ORACLES data archive (ORACLES Science Team, 2017). The instrument is calibrated before and after each deployment using the Langley plot technique (Schmid and Wehrli, 1995); in addition, corrections for non-uniform azimuthal dependence of the transmission of the optical fiber path were assessed after each flight calibration (Dunagan et al., 2013) and corrected for in post-processing, resulting in average



AOD uncertainty of 0.011 at 500 nm (LeBlanc et al., 2019). 4STAR also provides other quantities, for example, column water vapor and trace gas retrievals, which are not used here. HSRL-2 is a downward pointing lidar that provides vertical profiles of aerosol backscatter and depolarization at 355 nm, 532 nm, and 1064 nm wavelengths. Aerosol extinction is measured at 355 nm and 532 nm wavelengths (Hair et al., 2008; Burton et al., 2018). When the ER-2 was collocated with the P-3, imagery from the eMAS multispectral imager (King et al., 1996; Ellis et al., 2011) provided scene context.

## 2.4 Methods of Sampling: Radiation Walls and Spirals

There are two ways to determine aerosol intensive optical properties from irradiance and AOD. An algorithm by Schmidt et al. (2010a) uses nadir and zenith irradiance pairs above and below a layer to retrieve SSA, g, and the surface albedo. A different algorithm, by Bergstrom et al. (2010), first derives the layer absorption and scene albedo from the irradiance pairs above and below the layer, and then infers SSA, assuming a fixed value for g. Both methods were applied to clear sky and require irradiance measurements above and below a layer along with the associated AOD, which are most often obtained from individual points along the upper or lower leg of a "radiation wall" as shown in Figure 1.

The intent of the wall is to obtain scene albedo, layer absorption or transmittance by bracketing the aerosol layer above and below when flying at multiple altitudes along a track of about 100 km length. When only one aircraft is available, it samples the required legs sequentially, taking over an hour to complete. In clear sky, an aerosol layer will likely not change substantially during this time. However, in cloudy skies such as those encountered during ORACLES, the time lag between sequential sampling of the upper and lower leg is large enough that the cloud field is likely to change. Figure 1 illustrates the sampling for only two altitudes: at the bottom of the layer (BOL) and at the top of the layer (TOL) of interest. In the case of ORACLES, the BOL leg is located just below the aerosol layer and just above the cloud layer, where the column AOD and scene albedo are measured. The TOL layer is above the aerosol layer and the cloud, from which HSRL-2 measures profiles of extinction. Many other legs, for example below and within the cloud, and within the aerosol layer, were typically flown in addition to the BOL and TOL legs.





The net irradiance ($F_\lambda^{net}$) at any level is the difference between the downwelling and upwelling irradiance. The absorption $A_\lambda$ of a layer can be determined from the difference of the net irradiance at the upper and the lower boundary (the vertical component $V_\lambda$ of the flux divergence) if the horizontal flux divergence of radiation $H_\lambda$ is negligible ($|H_\lambda| \ll |V_\lambda|$). Under horizontally homogeneous conditions, we assume $A_\lambda = V_\lambda$, which is usually the case, giving:

$$A_\lambda = V_\lambda = \frac{(F_{\lambda,tol}^{net} - F_{\lambda,bol}^{net})}{F_{\lambda,tol}^{\downarrow}} = \frac{[(F_{\lambda,tol}^{\downarrow} - F_{\lambda,tol}^{\uparrow}) - (F_{\lambda,bol}^{\downarrow} - F_{\lambda,bol}^{\uparrow})]}{F_{\lambda,tol}^{\downarrow}} \,, \tag{1}$$

where $A_\lambda$ and $V_\lambda$ have been normalized by the incident irradiance at the top of the layer ($F_{\lambda,tol}^{\downarrow}$).

Under partially cloudy conditions, Schmidt et al. (2010b) found that $H_\lambda$ of a cloud layer is no longer negligible and can attain a magnitude comparable to $A_\lambda$ itself. Song et al. (2016) described the physical mechanism and spectral dependence of $H_\lambda$, which was determined by bracketing the cloud layer with irradiance measurements. For ORACLES, the aerosol above clouds, rather than the cloud itself, constitutes the layer of interest, but Figure 1 illustrates how non-zero values of $H_\lambda$ may arise under inhomogeneous conditions. The aerosol retrievals are only accurate if $|H_\lambda| \ll |V_\lambda|$, which ensures that Eq. (1) holds. The data analysis showed that this condition was rarely met for wall measurements, but more often during spiral measurements (section 3.1)

The radiation spiral, shown in Figure 2 and illustrated conceptually in Figure 1, provides multiple irradiance and AOD samples throughout the layer. Such a sampling pattern provides irradiance measurements at four headings throughout the column at a high vertical resolution without increasing the duration of the profile because the aircraft keeps descending or ascending during the straight segments, typically at 1000 feet per minute. During ORACLES, spirals were on average completed over the course of 10-20 minutes, depending on the vertical extent of the aerosol layer. Since typical roll angles during spirals were 15-30°, exceeding the operating range of ALP, the spirals include short, straight segments of 20-30 seconds duration every 90° heading change, which leads to a rounded square shaped pattern as shown in Figure 2b and 2d. SSFR acquires the irradiance profile over a minimal horizontal extent approximately 10 km in both latitude and longitude, reducing cloud and aerosol inhomogeneity effects, and over a much shorter time interval relative to the wall, maintaining correlation of measured irradiances throughout the spiral to the ambient cloud field. Moreover, the four

heading angles allow diagnosing biases from mechanical mounting offsets of ALP or reflections and obscuration by the aircraft structure. Acquiring a large number of samples over a relatively limited horizontal extent also reduces the impact of cloud albedo variability on the nadir irradiance. The downside of the spiral sampling is that it does not capture the spatial variability of the scene albedo,

which is assessed by the radiation wall. Therefore, in order to investigate any spatial relationships between radiative effects and albedo, spiral measurements must be used in conjunction with AOD and scene albedo measurements from the radiation wall where the albedo is defined as:

$$albedo_\lambda = \frac{F_\lambda^\uparrow}{F_\lambda^\downarrow} \; . \tag{2}$$

**2.5 Case Selection**

To characterize the connections between DARE, aerosol properties, and scene albedo, we chose to explore cases based on a) the availability of measurements from both a radiation wall and a radiation spiral, b) relatively high aerosol loadings, and c) a range of measured albedos. The first case is 20 September 2016, which had an AOD of 0.57 at 501 nm measured just above clouds during the spiral, with cloud albedos from the BOL leg of the radiation wall ranging from 0.39 to 0.59 at 501 nm. The

radiation spiral was located at the northern end of the BOL leg. The ER-2 flew in coordination with the P-3, such that eMAS imagery is available for context. Figure 2b shows an eMAS image overlaid with the flight track of the P-3 for the spiral flight pattern, along with the ER-2 flight track. The second case on 13 August 2017 was chosen because of the different cloud structures encountered along the BOL leg of the radiation wall. We treat this leg of the radiation wall as two separate cases based on differing

albedo ranges – the northern end of the wall, where the albedo values at 501 nm range from 0.06 to 0.39, and the southern end of the wall, where the albedo at 501 nm ranges from 0.29 to 0.75. The radiation spiral was located on the southernmost point of the southern case, though we utilize the retrieval results for both the north case and the south case DARE calculations. Figure 2d shows the spiral flight path overlaid on visible imagery from SEVIRI (Spinning Enhanced Visible and Infrared

Imager) onboard the geostationary Meteosat Second Generation (MSG) (Schmid, 2000). The AOD at 501 nm measured just above clouds during the spiral was 0.22. Table 1 lists the important parameter



ranges for both spirals: UTC, latitude, longitude, solar zenith angle (SZA) and albedo. Table 2 lists these parameters for the BOL legs for each of the three cases.

## 3 Methods

Our method to derive DARE from the observations is done with minimal assumptions. The DARE calculation is directly tied to the measured irradiances above and below the aerosol layer, and the AOD measured below the layer, since SSA and g by definition are consistent with these measurements. This differs from derivations from a) in-situ observations where the aerosol properties are de-coupled from the radiation fields and b) remote sensing observations where SSA and g are often prescribed based on an aerosol parameterization by type or region. By ensuring that SSA and g are consistent with the irradiance measurements in our approach, such assumptions are minimized when deriving DARE.

### 3.1 Irradiance Measurements: Walls vs. Spirals

To derive accurate aerosol absorptance from SSFR measurements, irradiance pairs above and below the layer must first be obtained. For the radiation walls, the irradiance pairs are sampled from the BOL and TOL legs at coincident locations, neglecting cloud advection and cloud evolution during the elapsed time between the two. For spirals, the entire measurement profile from above cloud to above aerosol is used to establish a linear fit of the data from which irradiance pairs are derived, improving the sampling statistics compared to radiation wall irradiance pairs. Figure 3 illustrates SSFR measured nadir and zenith irradiances for aircraft attitudes within the operating range of ALP plotted against the 4STAR AOD at 532 nm as vertical coordinate. Uncertainty bars are included for a subset of the measurements. Prior to fitting, all data are corrected to the SZA at the midpoint of the spiral to account for the minor change in solar position throughout the spiral:

$$F_\lambda = F_\lambda * \frac{\mu_0}{\mu},\qquad(3)$$

where $\mu = \cos(SZA)$ and $\mu_0$ is the value at the midpoint of the spiral.

To derive the BOL and TOL irradiances from the spiral using all the data, a linear regression is performed:





$$F_\lambda^\uparrow = a_\lambda^\uparrow + b_\lambda^\uparrow * AOD_{532}, \tag{4a}$$

$$F_\lambda^\downarrow = a_\lambda^\downarrow + b_\lambda^\downarrow * AOD_{532}, \tag{4b}$$

where $a_\lambda$ and $b_\lambda$ are the slope and intercept of the linear fit lines. The data points from the spiral are used collectively to establish the fit coefficients in (4a,b), which express the change of nadir and zenith

irradiance with AOD. Subsequently, the irradiance values at BOL and TOL are determined from $AOD_{532}^{max}$, measured at the bottom of the layer, and $AOD_{532}^{min}$, measured at the TOL. This method is more robust than picking individual irradiance pairs from the wall because many more data points are used. The uncertainty of the fit coefficients is dominated by the variability of the data throughout the vertical profile, rather than by the radiometric uncertainty of the contributing data points, discussed in

more detail in Section 3.4.

At a wavelength with small aerosol effects, such as 1.6 μm, shown in Figure 3b,d, neither nadir nor zenith irradiance change significantly as the aircraft moves through the layer. Any observed non-linearity or variability in the vertical profile at this wavelength can be ascribed to spurious measurement errors: for zenith, these could be due to reflections or obstructions by the aircraft or other factors

causing a transient variability in the downwelling irradiance. For nadir, this is attributed to albedo changes of the cloud field below. By contrast, the irradiance at 532 nm (Figure 3a,c) changes considerably throughout the vertical profile. The zenith irradiance decreases with decreasing altitude due to the increasing attenuation by the aerosol layer. The nadir irradiance shows the opposite behavior, decreasing with *increasing* altitude. By comparison, zenith and nadir irradiance would change in

lockstep for a purely scattering layer because the net irradiance remains constant in the absence of absorption.

To ensure that the aerosol signal is isolated from that of the variability of the underlying scene and that the data quality is sufficient to produce reasonable retrievals of SSA and g, a series of data filtering steps are applied.

(1) Filter the data in altitude to encompass only the aerosol layer. This ensures a maximum change in the irradiance during the vertical profile with minimum signal variations due to horizontal changes in the cloud field underneath (nadir) or any variability in the zenith signal unrelated to the aerosol layer.



Figure 2a,c show the spiral data as a function of altitude, with color coding to highlight data that passes the altitude filter.

(2) Select a subset of nadir data to focus on either predominantly clear or cloudy regions within the geographical footprint of the spiral. For the 20 September 2016 spiral, we focused on the cloudy pixels by selecting a longitude range of 8.86°E to 8.98°E based on the eMAS imagery, illustrated in Figure 2b, thereby eliminating regions that were substantially darker than the rest of the scene. The 13 August 2017 spiral did not require this filter because there were no clear regions distinguishable from cloudy regions (Figure 2d).

(3) Exclude data points where the nadir irradiance at 1.6 μm exceeds one standard deviation of the mean. These points are rejected to minimize the impact of cloud spatial inhomogeneity on the upwelling signal. Figure 3 indicates for each case the points that are included in the zenith and nadir linear fits and those that are outside of the standard deviation limit. The data points that are outside of the altitude and geographic filters are not shown. The aerosol loading on 13 August 2017 was significantly lower than on 20 September 2016, as well as the number of valid SSFR data points. This case was specifically chosen to explore the feasibility and sensitivity of the retrieval to variability in the upwelling irradiances and aerosol loading.

### 3.1.1 Horizontal Flux Divergence

Having obtained irradiance pairs from the wall or the spiral, the next step is to ensure that $|H_\lambda| << |V_\lambda|$ to minimize the impact of horizontal flux divergence in the subsequent retrieval of aerosol intensive optical properties. At long wavelengths, $H_\lambda$ asymptotically approaches a constant value as described by Song et al. (2016), which we denote as $H_\infty$. At the same time, aerosol absorption decreases with increasing wavelength (and thus decreasing optical thickness). Figure 4a shows $V_\lambda$ plotted as a function of $AOD_\lambda$ for 20 September 2016 and 13 August 2017. The intercept at $AOD=0$ ($A=0$ by definition) determines $H_\lambda$ because any non-zero measurement of $V_\lambda$ must originate from $H_\lambda$ in the absence of absorption. In the limit of $\lambda \to \infty$:

$$lim_{AOD(\lambda) \to 0} V_\lambda \equiv H_\infty . \qquad (5)$$





Thus, even though we do not determine $H_\lambda$ directly, $H_\infty$ is straightforward to obtain. Because of the findings of Song et al. (2016), $H_\lambda$ is zero for all wavelengths if $H_\infty$ is zero. Therefore, it is justified to apply Equation (1) to estimate $A_\lambda$ only if $H_\infty = 0$.

Tables 3a and 3b show that the calculated $H_\infty$ values for the filtered spiral data are near zero, but
significantly higher for the walls. For the 2016 case, $H_\infty$<0.2% for the spiral, and up to 15% for the irradiance samples from the wall. $H_\infty$ is larger for the 13 August 2017 spiral, about 2%, which could be due to the larger scene inhomogeneity based on the available imagery. It makes sense that the wall measurements have larger values for $H_\infty$, mainly because the collocated pairs do not necessarily represent the irradiance of the same scene, considering the time difference between the BOL and TOL
legs. In addition, the effective footprint of the nadir SSFR light collector (the circle from within which half of the signal originates) changes at different altitudes, which means that the horizontal extent of cloud that contributes to the sampled signal for the TOL leg is much greater than for the BOL leg. While this is also true for the spiral, the standard deviation filtering effectively separates the aerosol signal from that of changes in scene albedo, including those due to the changing footprint size of SSFR
with altitude.

To quantify the horizontal variability in the flux field relative to the aerosol absorption, we introduce the inhomogeneity ratio

$$i_\lambda = \frac{H_\infty}{V_\lambda - H_\infty} . \tag{6}$$

The denominator approximates the true absorption, where the horizontal flux contribution to the
observed $V_\lambda$ has been subtracted to yield $A_\lambda$ (though we have substituted $H_\infty$ for $H_\lambda$). $V_\lambda$ and $H_\infty$ are both measurable quantities, while $A_\lambda$ can only be inferred from $V_\lambda$ if $H_\infty$ is near zero. If $H_\infty$ is similar (or exceeds) in magnitude to $V_\lambda$, $H_\lambda$ will also be of similar magnitude, and we cannot determine $A_\lambda$ from $V_\lambda$. The spectral inhomogeneity metric provides an empirical method to determine when this occurs.

Table 4 summarizes the interpretation of $i_\lambda$ values, which can be either positive or negative due to the
horizontal flux divergence; when $i_\lambda$ is positive, this indicates a divergence of radiation within the layer (apparent absorption), and when $i_\lambda$ is negative it indicates a convergence (apparent emission). We expect that as the wavelength becomes longer, the magnitude of $i_\lambda$ will also increase since the aerosol



absorption is largest at the shortest wavelengths, while $H_\lambda$ is not strongly wavelength dependent. Tables 3a and 3b list the $i_\lambda$ values at 355, 532, and 1650 nm for the spiral and, for illustration, the maximum and minimum $i_\lambda$ values from the radiation walls. Both spirals exhibit near zero $i_\lambda$ values at 355 and 532 nm, though the 13 August 2017 values are slightly closer to 1 in large part due to the lower aerosol

loading, and the retrieval from 20 September 2016 is therefore more reliable than 13 August 2017. The maximum (minimum) $i_\lambda$ values for the radiation walls are larger (smaller) than the spiral values at all wavelengths. The specific $i_\lambda$ values for which performing an aerosol retrieval is minimally affected by $H_\lambda$ are subjective, and a follow up paper will further develop and characterize the limits by investigating more cases from ORACLES.

Because of the high $H_\infty$ and $i_\lambda$ values, the wall measurements are not used to determine aerosol absorptance or for the SSA and g derivation. Conversely, the near zero $H_\infty$ values and low $i_\lambda$ values of the spirals allow us to substitute Equations 4a and 4b into Equation 1, which simplifies to:

$$A_\lambda = \frac{AOD_{532}^{max} * (b_\lambda^\uparrow - b_\lambda^\downarrow)}{a_\lambda^\downarrow} \tag{7}$$

The spiral-derived absorptance spectra for a) 20 September 2016 and b) 13 August 2017 are shown in
Figure 4b. The largest absorptance occurs in the water vapor bands of 1870 nm, 1380 nm, 1100 nm, and 940 nm. In the relatively water-free spectral range, approximately 900nm and shorter, the absorptance is dominated by aerosol absorption (except for a few water vapor bands with relatively low absorption, the Oxygen A- and B-bands, the Chappuis ozone absorption band, and other trace gas absorption). The 4STAR AOD retrieval wavelengths specifically avoid the gas absorption features, although those that
coincide with the Chappuis ozone absorption band and other trace gas absorption bands are unavoidable, and are accounted for in the 4STAR retrieval (See Appendix of LeBlanc et al., 2019).

The subsequent retrievals of SSA and g use the individual upwelling and downwelling irradiances rather than the absorptance from the spiral profiles. Lacking other constraints, we assume that since $A_\lambda$ is unaffected by cloud inhomogeneity when $H_\infty$ is near zero, the same is true for the irradiances from
which $V_\lambda$ is originally calculated. $H_\infty$ and $i_\lambda$ serve as metrics to assess the suitability of data for the aerosol retrieval.





## 3.2 SSA Retrieval

The retrieval of SSA and g is done with the publicly available 1-dimensional (1D) radiative transfer model (RTM) DISORT 2.0 (Stamnes et al., 2000) with SBDART for atmospheric molecular absorption (Ricchiazzi et al., 1998) along with the standard tropical atmosphere available within the public library

libRadtran *(*Emde et al., 2016; libradtran.org*)*. In contrast to the algorithms by Pilewskie et al. (2003), Bergstrom et al. (2007) and Schmidt et al. (2010a), the aerosol layer is located over a variable cloud scene, but otherwise the principle is the same. This work is most similar to the algorithm introduced by Schmidt et al. (2010a), for which SSA and g are retrieved simultaneously.

The RTM allows us to calculate upwelling and downwelling fluxes determined by inputs of the surface

albedo and the aerosol properties of AOD, SSA and the asymmetry parameter. The updated retrieval algorithm is based on the comparison between the calculated fluxes and the SSFR measured fluxes. Spectral albedo from SSFR and AOD from 4STAR are used as inputs, which leaves SSA and g as the free retrieval parameters. For 20 September 2016, the SZA within the RTM is set to 21.0 and the albedo at 501 nm is 0.45 while for 13 August 2017, the SZA is set to 33.5 and the albedo at 501 nm is 0.70.

Since the cloud albedo is directly measured, cloud properties such as COT and effective radius are not required – an advantage when compared to the associated remote sensing bias when obtaining it from space-borne imagery (Chen et al., 2019).

The first step in the retrieval is to condition the 4STAR AOD so that the AOD profile decreases monotonically with altitude. Because 4STAR samples horizontal as well as vertical variability

throughout the spiral, the AOD profile can sometimes deviate from a strictly monotonic decrease, which cannot be ingested by the RTM. We alleviate this problem by smoothing the AOD profile with a polynomial to eliminate minor deviations from monotonic behavior. For instances when the derived extinction becomes negative, we set the value to 0. Figure 5a (20 September 2016) and b (13 August 2017) visualize the original AOD profile and the corresponding polynomial. The unique altitude to

AOD relationship is used to derive the extinction profile, also shown in Figure 5a,b. Above the aerosol layer, any remaining AOD measured by 4STAR is assigned to a layer extending to 15,000 m (a top altitude chosen somewhat arbitrarily lacking the knowledge of the correct height distribution of the residual AOD).





In the second step of the retrieval, the RTM calculates the upwelling and downwelling irradiance profiles for a given pair of {SSA, g}. The modeled downwelling irradiance profile is rescaled such that the model results at the TOL are consistent with the measured downwelling irradiance. The scaling factor effectively allows for inaccurate values in the extraterrestrial solar flux (Kurucz, 1992),
differences in atmospheric constituents, such as aerosols above the aircraft's top altitude, or for absorbing gases not accounted for using the standard atmospheric profile. It is typically close to 1. At the BOL, the measured upwelling irradiances are also rescaled such that the model albedo is consistent with measured albedo. If the calibration for the upwelling and downwelling irradiance is consistent, the scale factors should be the same. Therefore, any retrieval with differing nadir and zenith scale factors is
flagged as failed.

The third step of the retrieval determines the most probable pair of {SSA,g} and calculates the uncertainty. For each {SSA,g} pair calculation, every SSFR data point in the profile is assigned a probability according to the difference between the calculation and the measurement. The probability of {SSA, g} given the SSFR observations is determined from the Gaussian distribution that represents the
measurement uncertainty. This is illustrated in Figure 6a. The probability of that pair given the observations is determined by multiplying the individual probabilities within the profile. The {SSA,g} pair with the highest probability value is reported as the retrieval result. The {SSA, g} pair probabilities are shown as a 2D probability density function (PDF) in Figure 6b, where the error bars show the 1-sigma uncertainty for SSA and g separately, determined by the respective marginal (1D) PDFs. Since
only the SSFR uncertainty is considered within the retrieval, the 4STAR uncertainty is treated separately by performing the retrieval three times: (1) for the nominal AOD, (2) for the nominal AOD – range of uncertainty, (3) for the nominal AOD + range of uncertainty. Figure 6b shows an example retrieval at 501 nm for the three retrievals. Finally, the retrieved spectra of 4STAR wavelengths between 355 nm and 660 nm of SSA and g are reported, with a range of uncertainty that encompasses
the three separate retrievals.

Currently, the retrieval is performed for each wavelength individually, and no spectral smoothness constraints are applied. This is an important difference compared to other methods such as the


AERONET inversion method that retrieves aerosol size distributions and the real and imaginary parts of the index of refraction for various size modes (Dubovik and King, 2000).

The retrieval also allows us to calculate the Absorption Ångström Exponent (AAE) from the absorbing aerosol optical depth (AAOD) which, like SSA, quantifies the radiative effects and optical properties of absorbing aerosols (Pilewskie et al., 2003; Bergstrom et al., 2010). The AAE and aerosol absorption optical depth AAOD are determined as follows:

$$AAOD=(1-SSA)*AOD, \tag{8a}$$

$$AAOD = AAOD_{500} * \left(\frac{\lambda}{\lambda_{500}}\right)^{-AAE}. \tag{8b}$$

We compare the AAE and SSA results from our retrieval to in situ measurements from a three-wavelength nephelometer (TSI 3563) and a three-wavelength particle soot absorption photometer (PSAP) (Radiance Research). The PSAP provides AAE, while the combination of scattering from the nephelometer and absorption from the PSAP provide SSA. Average values of SSA are weighted by the extinction, specifically to obtain a column value of SSA from the spiral profiles.

## 3.3 DARE and Critical Albedo

We calculate the DARE at the TOL and BOL as the difference between the net irradiance with and without the aerosol layer:

$$DARE_\lambda = F_{\lambda,aer}^{net} - F_{\lambda,no\ aer}^{net} \tag{9}$$

The DARE calculations are performed with the intensive aerosol properties (SSA and g) from the spiral profiles, and with the albedo as measured by SSFR from a BOL leg, AOD from 4STAR on the same leg, and HSRL-2 extinction profiles from the TOL leg to capture any variability within the aerosol encountered along the wall.

Combining vertical and horizontal sampling in this way is predicated on the assumption that the aerosol intensive properties do not change along the BOL leg, whereas albedo and AOD are expected to vary. This is a reasonable assumption as long as the legs do not cross an air mass boundary; in situ measurements show that along the BOL leg on 20 September 2016, the SSA at 530 nm ranges from 0.80 to 0.86 (0.83±0.01, average ± standard deviation). During this same time, the PSAP instrument





shows the AAE ranges from 1.71 to 2.02 (1.87 ±0.05). From a radiation wall leg within the aerosol layer (12:35-12:47 UTC), the SSA ranges from 0.84 to 0.87 (0.85 ± 0.004). The AAE ranges from 1.71 to 1.99 for this time (1.84 ± 0.04). On 13 August (both north and south sections), the SSA from the BOL leg ranges from 0.84 to 0.93 (0.87 ± 0.02) while the AAE from 0.97 to 2.1 (1.6 ± 0.3). Within the

aerosol layer (14:08-14:18 UTC), the SSA ranges from 0.88 to 0.90 (0.89± 0.003) and the AAE ranges from 1.80 to 2.16 (1.92± 0.07) (Dobracki et al., 2019).

Since the spectral information is available, we choose to calculate DARE spectrally (350 nm- 660 nm) as a percentage of the incoming radiation rather than as broadband values commonly reported. Within the RTM, the SZA is fixed to the mean value of the above cloud leg; for consistency, SSFR

measurements are corrected to this SZA following equation 3 (17.9° for 20 September 2016, 22.1° for northern case of 13 August 2017, and 23.2° for the southern case of 13 August 2017). The albedo ranges for each case are presented in Table 2, and the aerosol intensive properties used are presented in Figure 7. Although 0 and 1 albedo values were not actually encountered, we include them in the RTM runs and calculate the DARE to investigate the behavior at the albedo limits. The relationship between

DARE and SSFR measured albedo is nearly linear; therefore, we fit a line to the 1D calculations to find the x-intercept, which is the critical albedo.

To estimate the total DARE uncertainty, we combine the errors of the individual components:

$$\delta DARE_{total=} = \sqrt{\left(\delta DARE_g\right)^2 + (\delta DARE_{albedo})^2 + (\delta DARE_{AOD})^2 + (\delta DARE_{SSA})^2} \,, \qquad (10)$$

where each parameter uncertainty is calculated as:

$$\delta DARE_g = \frac{|DARE_{g+\delta g} - DARE_{g-\delta g}|}{2} \,, \qquad (11a)$$

$$\delta DARE_{albedo} = \frac{|DARE_{albedo+\delta albedo} - DARE_{albedo-\delta albedo}|}{2} \,, \qquad (11b)$$

$$\delta DARE_{AOD} = \frac{|DARE_{AOD+\delta AOD} - DARE_{AOD-\delta AOD}|}{2} \,, \qquad (11c)$$

$$\delta DARE_{SSA} = \frac{|DARE_{SSA+\delta SSA} - DARE_{SSA-\delta SSA}|}{2} \,. \qquad (11d)$$

The uncertainty of g and SSA are obtained from their retrieval, and the AOD uncertainty is the

measurement uncertainty. Since the albedo is a ratio of upwelling and downwelling irradiance,





calibrated using the same apparatus, the relative precision of the measurement to each other drives the uncertainty rather than through error propagation of each calibrated accuracy. The albedo uncertainty is estimated to be approximately 1%.

This method assumes all four individual uncertainties are uncorrelated, and most likely overestimates

the DARE uncertainty.

## 4 Results and Discussion

## 4.1 Aerosol Properties

The SSA spectra from 355 nm to 660 nm retrieved from the radiation spirals for each case are shown in Figure 7 a,b, and Table 5 presents a comparison between SSFR-derived SSA and AAE with past results

and in situ measurements from ORACLES. The 20 September 2016 case can be considered spectrally flat with a minimum SSA value of 0.83 (± 0.02) at 660 nm and a maximum SSA value of 0.86 (± 0.01) at 380 nm. The 13 August 2017 case shows a spectrally flat SSA with 0.83 (± 0.04) at 355 nm and 0.82 (± 0.07) at 660 nm. Compared to the SAFARI 2000 campaign results shown in Russell et al., 2010, the results from the two ORACLES cases are slightly lower: 0.87 at 501 nm compared to 0.85 (± 0.01) (20

September 2016) and 0.82 (± 0.05) (13 August 2017) at 501 nm, although the values are similar to those presented in Giles et al., 2012 for AERONET sites that experienced smoke aerosol events (Giles et al., 2002; Eck et al., (2003a, 2003b)). In situ measurements of the extinction-weighted SSA from the spiral profiles and are shown in Figure 7a,b. At 530 nm, the 20 September 2016 spiral had an average SSA of 0.86 with a standard deviation of 0.03, while the 13 August 2017 spiral had an average SSA of 0.88

with a standard deviation of 0.01. Table 4 presents a comparison at 500 and 530 nm between SSFR-derived SSA and AAE with past results and in situ measurements from ORACLES, and a detailed SSA inter-comparison can be found in Pistone et al. (2019).

Also included in Figure 7a,b are the uncertainty estimates for each wavelength, shown as the smaller, blue error bars. The larger, black error bars illustrate what the uncertainty would be if we had derived

the SSA using irradiance pairs rather than from the whole profile (i.e. if the spiral TOL and BOL values had been taken from a radiation wall.) The uncertainty derivation for the radiation wall measurements





requires the assumption that $H_\lambda=0$, though as we have shown this is not the case, and is described in detail in Appendix A. As can be seen in Figure 8, the uncertainty from the walls is much larger than from the new spiral method, and would be even larger if we included error due to $H_\lambda$.

Figure 7c shows the asymmetry parameter retrievals along with uncertainty estimates. The values (0.45-

0.65) for the 20 September 2016 case are within range of other estimates for the region, although the spectrum falls off more rapidly than assumed by Meyer et al. (2013). The large uncertainties for the 13 August 2017 case show that even for moderate mid-visible AOD (~0.3), the information content with respect to this retrieval parameter is fairly low. Despite the limited information content in the SSFR stand-alone retrievals, there is some indication that the asymmetry parameter always falls off more

rapidly than in previous assessments – with a value approaching zero for large wavelengths. This may be due to fewer coarse-mode aerosol particles than in previous climatologies for the region (Formenti et al., 2018).

The AAOD spectrum from which we derive AAE is shown in Figure 7d for both cases. The AAE for the 2016 case is 1.29 while the AAE for the 2017 case is 1.44. Both AAE values are similar to the

results of Bergstrom et al. (2007) and reproduced by Russell et al. (2010) from the SAFARI campaign for biomass smoke of 1.45 for wavelengths of 325 to 1000 nm. In situ measurements of AAE from the PSAP showed the average AAE values from the two spirals profiles to be 1.79 for 20 September 2016 and 1.70 13 August 2017 for the 470-660 nm wavelength range (Dobracki et al., 2019). Differences between radiatively-derived and in situ measured values for both AAE and SSA may due to differences

in aerosol humidification; the irradiances measured by SSFR and the resulting aerosol properties represent the aerosol in ambient conditions (Pistone et al., 2019). The in situ instruments, however, control the relative humidity while the aerosol is measured, potentially causing discrepancies. Biases may also be present in the in situ absorption that propagates to bias in SSA, due to known issues with measuring absorption on a filter (Pistone et al., 2019). When the aerosol intensive properties are derived

using our new approach, the aerosol optical properties are radiatively consistent with the measured irradiance and the ambient optical thickness, therefore allowing us to establish a more direct estimate of DARE.



## 4.2 DARE and Critical Albedo

Figure 8a shows the TOL radiative effect as percent of the incoming radiation at 501 nm as a function of the underlying albedo for 20 September 2016 and the north and south cases from 13 August 2017. Figure 8b shows example spectra from each case with associated error bars. A positive DARE value indicates that the aerosol warms the layer. For the 20 September 2016 case, the scene albedo, which we consider as the average of all the albedo values, is 0.5 at 501 nm with a corresponding TOL DARE of 9.6 ± 0.9 % (percentage of incoming irradiance). For the 13 August 2017 North case, the scene albedo of 0.03 results in a TOL DARE of -0.61 ± 2.01 % while the scene albedo of 0.27 for 13 August 2017 South results in a TOL DARE of 5.45 ± 1.92 %. As can be seen in Figure 8, the DARE from the 20 September 2016 case is larger than the 13 August 2016 cases, in large part due to the higher AOD values in the 20 September 2016 case. At the BOL, DARE is always negative since the amount of radiation reaching that altitude decreases when there is an aerosol layer present due to the scattering and absorption that occurs. For this reason, we do not show the BOL DARE results visually. At the scene albedos listed above, the BOL DARE values at 501 nm are -7.27 ± 0.9 %, -8.36 ± 2.01 %, -4.37 ± 1.92 %, for 20 September 2016, 13 August 2017 North, and 13 August 2017 South, respectively. For the 2017 cases, the clouds were broken on the north section and homogeneous on the south section. The TOL radiative effect crosses from negative to positive with increasing albedo, illustrating the same aerosol has a warming effect in the south and a cooling effect in the north due to the differences in the underlying cloud. This is similar to the conclusions of, Keil and Haywood (2003), Chand et al. (2009), and Meyer et al. (2013) who also find that DARE decreases as the underlying clouds darken, eventually becoming negative. We find that the critical albedo is 0.21 at 501 nm for 20 September 2016 and 0.26 for 13 August 2017. Chand et al. (2009), along with Meyer et al. (2013) and many other studies, choose to normalize the radiative effect by the aerosol optical depth, a quantity known as the radiative forcing efficiency (RFE), to isolate the cloud effect from the aerosol loading on DARE. For this region, Chand et al. (2009) find that the transition point from positive to negative RFE is at the critical cloud fraction of 0.4. Since we are interested in the radiative effects as a function of both the cloud and aerosol properties, we choose not to translate DARE into RFE since it a) removes the dependence on the aerosol loading and b) may not linearly scale with mid-visible AOD, with evidence suggesting that the





increase depends upon the cloud albedo (Cochrane et al., in prep). We can, however, convert critical albedo into critical cloud fraction and critical optical thickness. For a cloud fraction of 100% and using the two-stream approximation (Coakley and Chylek, 1975), a critical albedo of 0.21 (0.26) corresponds to a critical optical thickness of 1.5 (1.35). Assuming the mean cloud albedo value of 0.5 used by Chand

et al., 2009 (determined on the basis of Jul.–Oct. 5°×5° mean and standard deviation of MODIS-retrieved cloud optical depths), a critical albedo value of 0.21 (0.26) and g value of 0.56 (0.27) would translate into a critical cloud fraction of 0.42 (0.52). This is consistent with their finding of the critical cloud fraction to be 0.4. Podgorny and Ramanathan (2001), however, find a much lower critical cloud fraction even with a higher SSA. Chand et al. (2009) attributes this discrepancy to differences in cloud albedo,

acknowledging that accurate cloud albedo values are crucial in determining aerosol radiative effects. In reality, one cannot simply fix the cloud albedo to a single value; the true albedo at 501 nm for 20 September 2016 ranges from 0.42 to 0.60 while the 13 August 2016 albedo ranges from 0.01 to 0.4. Using critical albedo instead of critical cloud fraction or optical thickness circumvents these problems. Chand et al. (2009) find that the critical cloud fraction is particularly sensitive to the SSA and is the

greatest source of explicitly estimated uncertainty in their study. In our study, the largest uncertainty contributor to the DARE calculation and, similarly, critical albedo, is case dependent, though the SSA represents a significant fraction of the error across all cases for wavelengths of 355 nm-660 nm. This can be seen in Figure 9, which shows an example the uncertainty contributions of each input parameter to the DARE calculation for one point from each case. The 20 September 2016 DARE error is

dominated by the SSA, while the 13 August 2017 North case is dominated by the g error. The 13 August 2017 South case has nearly equally large contributions from SSA and g. It should be noted that the uncertainty partitioning changes for different points along the radiation wall. Quantifying the individual component uncertainties, especially the albedo uncertainty, is an advancement to satellite-based studies that focus on only quantifying the aerosol parameter uncertainties.  The uncertainty due to

the underlying clouds in DARE calculations, while known to be important, is often not emphasized or quantified since the cloud albedo cannot be measured directly from space. Despite the differences between previous studies and our work, the results all highlight the importance for accurate optical





properties of both the aerosol and underlying cloud layers, since the radiative effect of an aerosol layer so clearly depends on both.

## 5 Summary and Future Work

Aircraft observations, such as those taken during ORACLES, help capture some of the information

relevant for determining the aerosol radiative effect in the presence of clouds that satellite measurements are unable to obtain: aerosol SSA, g and cloud albedo. The aerosol properties, SZA, and albedo differed between the cases examined in this work, and the critical albedo was 0.21 for 20 September 2016 and 0.26 for 13 August 2017. The critical albedo parameter describes how a certain type of aerosol is affected by the underlying surface despite scene differences. If shown to be applicable

across many scenes, this parameter could be very useful for parameterizations of DARE above clouds for biomass burning aerosol.

DARE, by definition, requires radiative transfer modeling and our calculations utilize AOD from 4STAR, measured cloud albedo from SSFR, and retrieved values of SSA and g. Using SSFR irradiance measurements from a square spiral, which is made possible by SSFR in conjunction with ALP, turned

out to be crucial for determining aerosol intensive properties for the inhomogeneous or changing situations encountered during ORACLES. The newly developed retrieval algorithm allowed us to separate cloud effects from aerosol effects through filtering methods which account for a changing cloud field by eliminating regions of high variability and points that are subjected to 3D effects. We determine this through the $H$ parameter, a proxy for 3D cloud effects, which is near zero for the filtered

spiral measurements, but not for the "wall" measurements (stacked legs). The spiral method also considerably decreases the uncertainty on the retrieved SSA compared to the radiation wall method, which is of key importance since the SSA is largest contributor to the overall DARE uncertainty.

As expected, we found that DARE increases with AOD. However, upon examining other cases (Cochrane et al., in prep), evidence suggests that the increase is not linear with AOD and depends upon

the cloud albedo.  This puts into question the utility of the concept of radiative forcing efficiency that has been widely used in studies such as Pilewskie et al. (2003), and Bergstrom et al. (2003), Redemann et al. (2006), Chand et al. (2009), Schmidt et al. (2010a), and LeBlanc et al. (2012). Although these





references did not explicitly assume linearity, one must be cautious when using RFE to make the link from satellite-derived optical thickness to DARE. This provides motivation for developing a new approach for establishing such a link, for which the critical albedo could provide that connection as it accounts for both the aerosol and cloud properties.

Future work will also be aimed at verifying whether the DARE-albedo relationship found in this case study is generally valid across scenes with different cloud spatial inhomogeneities, different sun angles, etc. Work will also be aimed at assessing the remaining suitable ORACLES cases by applying the methodologies presented in this paper to determine regional values of SSA, g, DARE and heating rate profiles. The results will be used to parameterize the radiative effects in terms of appropriate quantities

such as the AAOD, and will be presented in a follow-up paper (Cochrane et al., in prep). It is also important that the SSA be checked for consistency with SSA retrieved from other instruments from the ORACLES campaign, an effort that is already underway (Pistone et al., 2019).

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

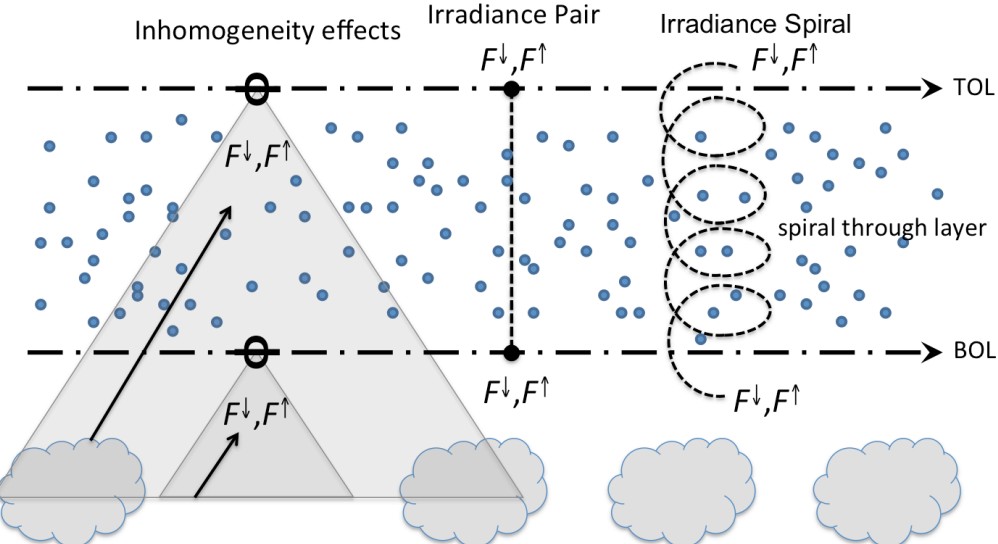

**Figure 1: Schematic of a radiation wall, radiation spiral, and the appearance of horizontal flux divergence ($H_\lambda$) in SSFR**
25      **measurements. During a radiation wall, SSFR measures upwelling and downwelling irradiance along the Top of Layer leg (TOL)**
**and Bottom of Layer (BOL) leg, which are collocated in space but not in time. During the radiation spiral, SSFR measures**





upwelling and downwelling irradiance throughout the entire aerosol layer. The left side of the figure illustrates an example of how non-zero $H_\lambda$ arises in SSFR measurements under certain cloud conditions. The gray triangles figuratively represent the viewing geometry of SSFR at the TOL and BOL. Ignoring any change in clouds over time, the TOL SSFR-measured irradiances include contributions from a larger area than at the BOL. Under inhomogeneous conditions, the TOL and BOL SSFR measurements contain differing cloud scenes; in our illustration, the BOL measurement has little to no signal contribution from clouds, whereas the TOL measurement has a large contribution of the signal from clouds. The upwelling irradiance at the TOL would therefore be larger (smaller net irradiance) than at the BOL (larger net irradiance) due to the bright clouds.

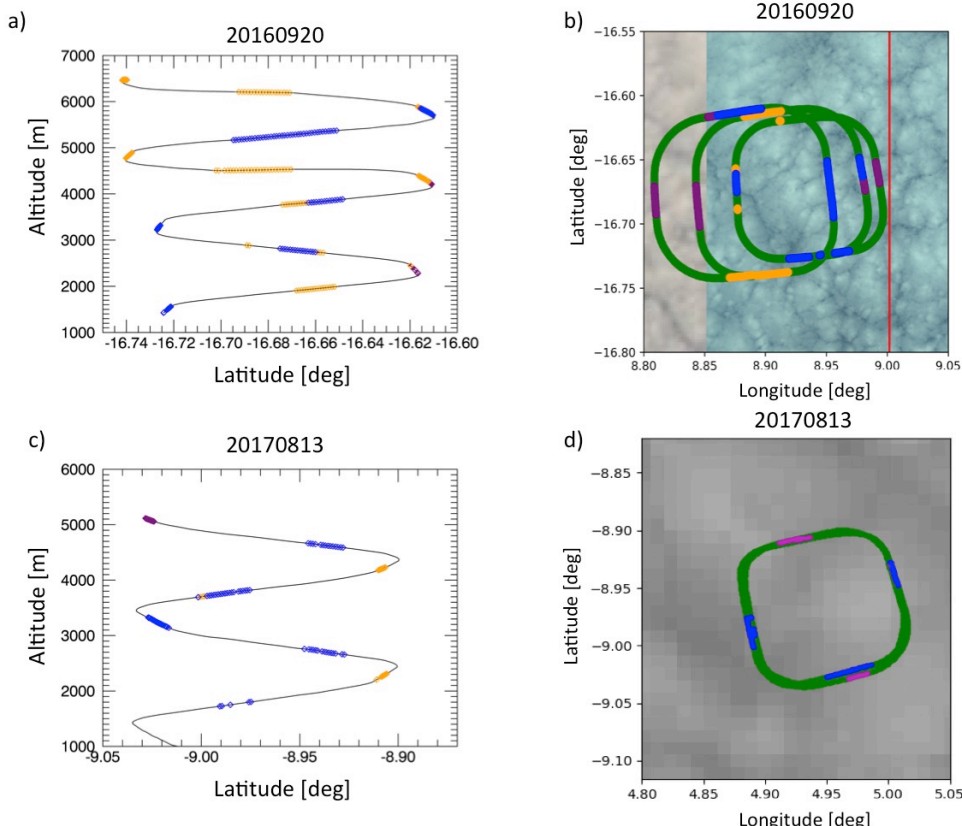

Figure 2: a) The latitude vs. altitude of filtered spiral data for 20 September 2016. b) The corresponding high-resolution eMAS imagery (blue) with lower resolution MODIS imagery (gray). Overlaid is the P-3 spiral flight track in green and ER-2 flight track in red. c) The latitude vs. altitude of filtered spiral data for 13 August 2017. d) Corresponding SEVIRI imagery. For 20 September 2016, the altitude range is 1.4 to 6.5 km while for 13 August 2017 the altitude range is 1.7 to 5 km. For all 4 figures, the purple color shows data that are within the limits of the ALP, but do not pass the geographic or the standard deviation filter. The orange color shows the data that have passed the geographic filter but do not pass the standard deviation filter. The blue points meet all of the requirements and are the data used within the linear fit to determine the TOL and BOL irradiances.




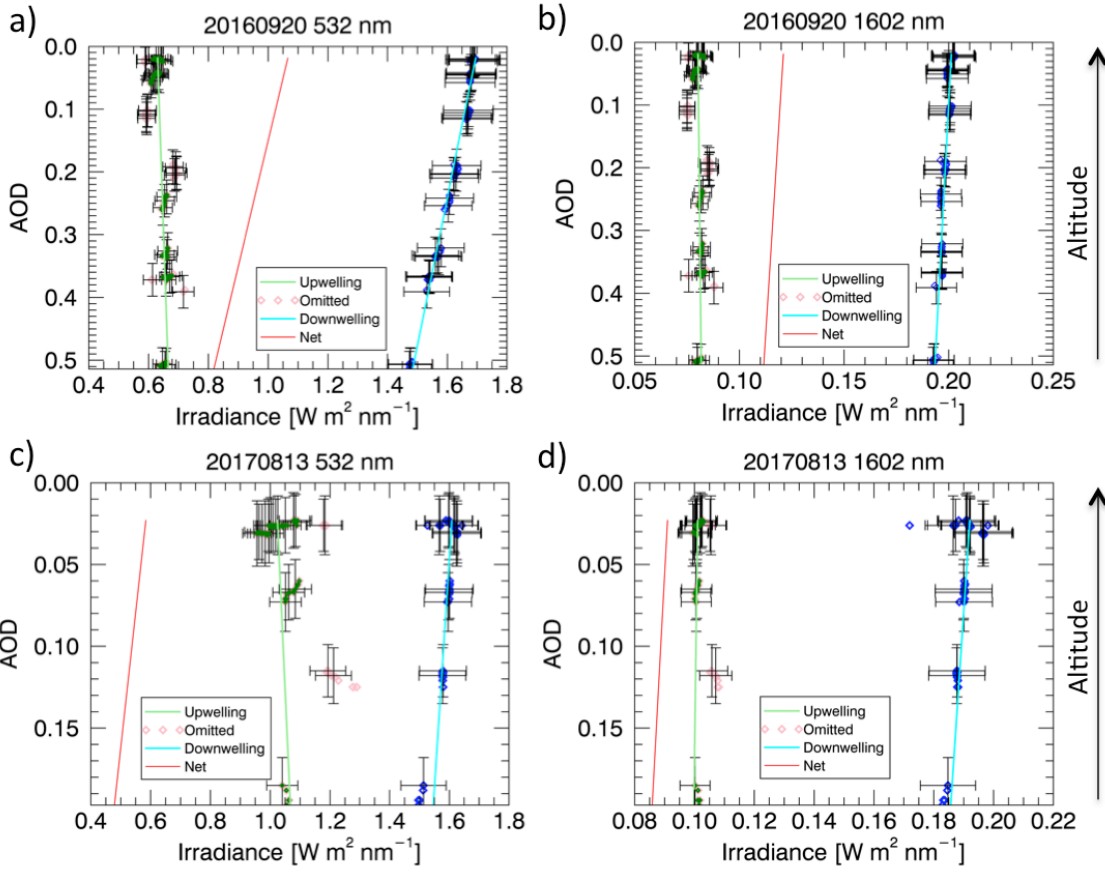

**Figure 3: Examples of the filtering and extrapolation technique for 20 September 2016 a) 532 nm and b) 1602 nm and for 13 August 2017 radiation spirals at c) 532 nm and d) 1602 nm. SSFR irradiance measurements are plotted against 4STAR above-aircraft AOD at 532 nm along with the associated measurement uncertainty. The omitted upwelling data (pink) did not pass the standard deviation or geographic filter and is not used for the calculation of the linear fit. All zenith measurements are included in the fit. At 1602 nm, there is little to no aerosol absorption and the net irradiance is expected to be nearly constant with altitude. At 532 nm however, there is aerosol absorption and the net irradiance decreases with increasing AOD.**





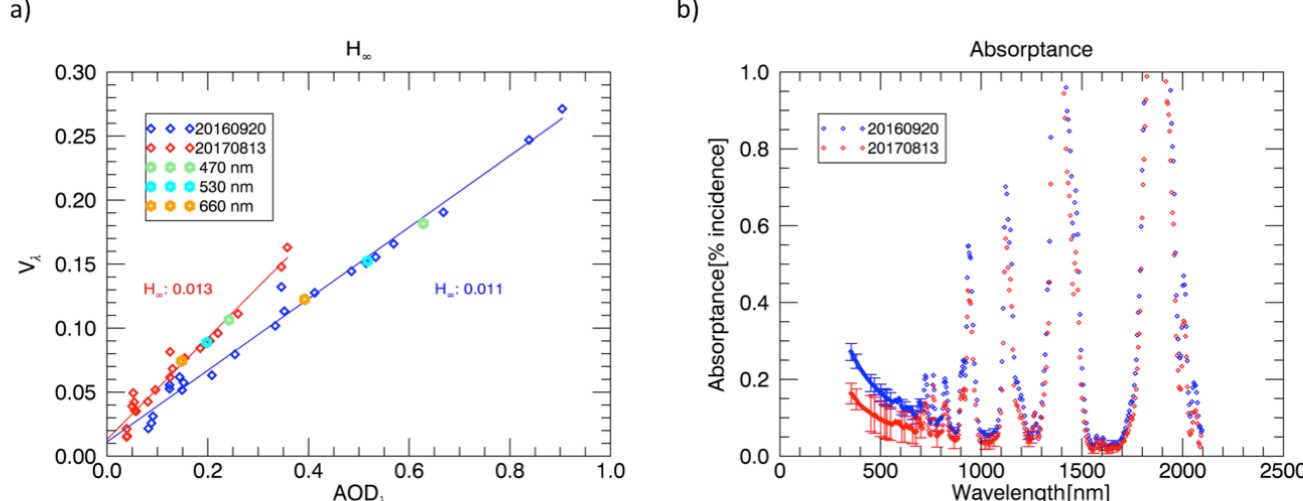

**Figure 4a: Examples of the method for determining $H_\infty$. At long wavelengths, the horizontal flux divergence, $H_\lambda$, asymptotes to a constant value ($H_\infty$); a non-zero value indicates 3D effects. Here we perform a linear fit between $AOD_{\lambda,max}$ and $V_\lambda$, the vertical flux divergence, for all 4STAR wavelengths where $H_\infty$ is the y-intercept, thereby bypassing the necessity of determining $H_\lambda$ directly. Figure 4b: The spiral derived for 20 September 2016 and 13 August 2017. Uncertainty estimates are shown as error bars at the 4STAR wavelengths.**




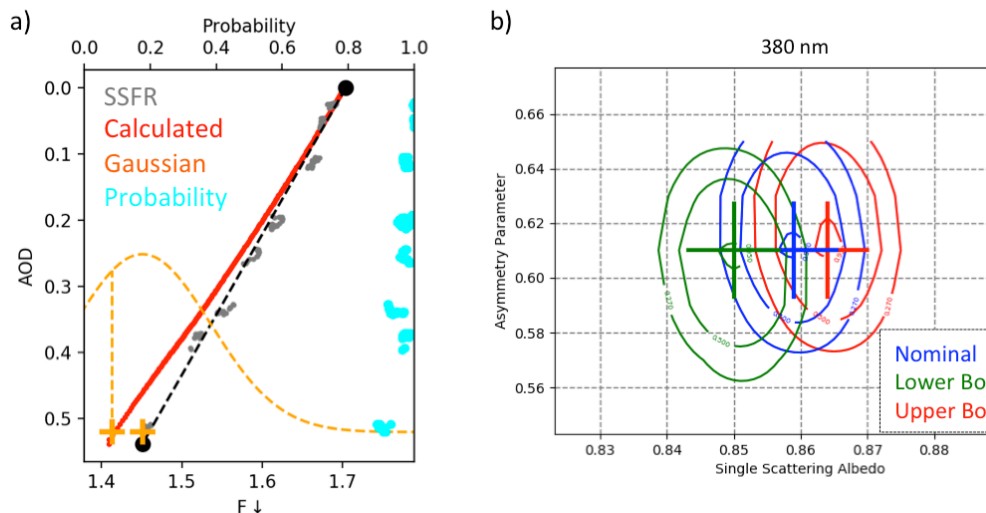

**Figure 6. a) This figure shows measurements of downwelling irradiance (grey) along with a calculated profile (red) for one pair of SSA and g. The probability of this pair, given the measurements, is obtained by considering the measurement uncertainty range (represented as a Gaussian, yellow) for the individual data points, and assigning a probability (cyan, upper axis) to each data point according to the difference between the calculation and the measurement. The individual probabilities are then multiplied throughout the profile and constitute the probability of the {SSA,g} pair given the observations. b) Shows these probabilities as a function of SSA and g, calculated for the nominal 4STAR AOD (blue) and for the upper (red) and lower (blue) bound of the reported uncertainty range. The ellipses represent confidence levels of 27%, 50%, and 95%.**




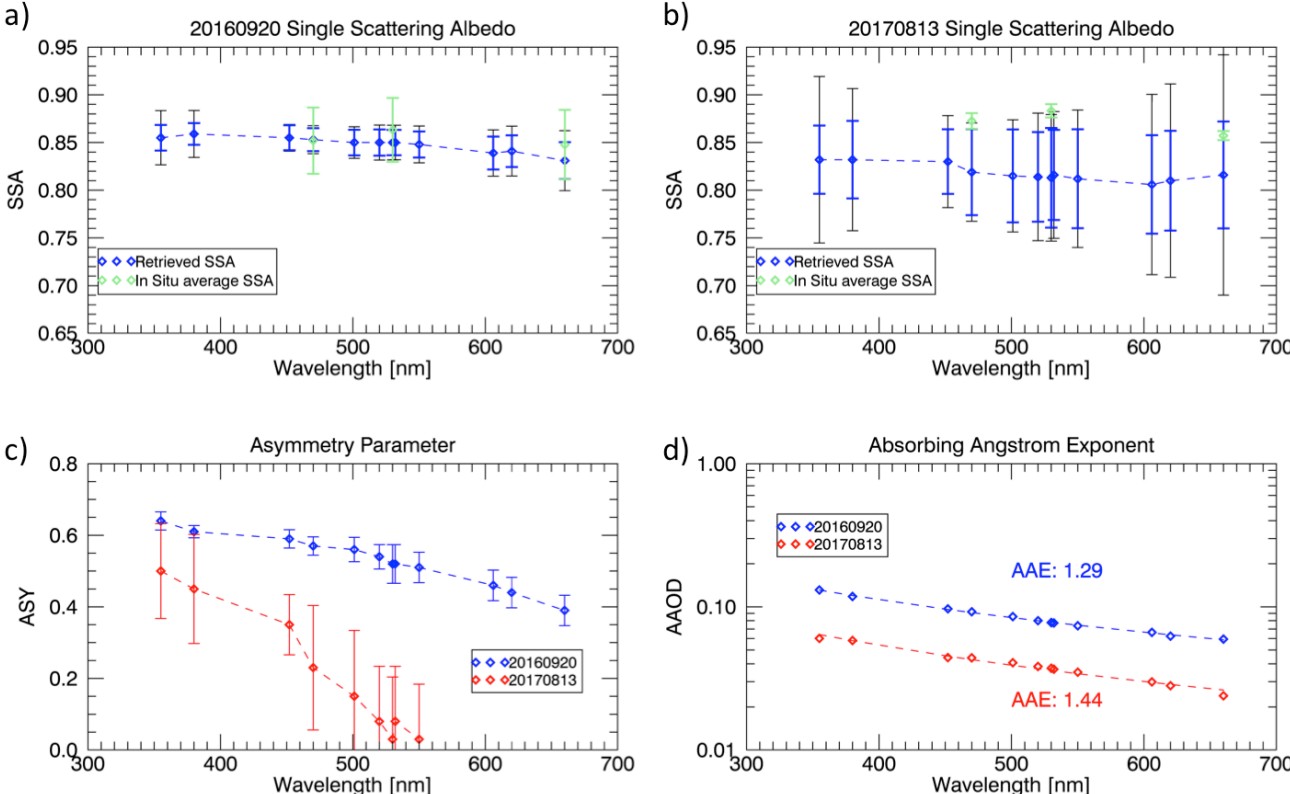

**Figure 7. Spiral derived SSA values for a) 20 September 2016 and b) 13 August 2017 with associated error bars. The smaller error bars in blue are the spiral uncertainty estimates; the larger error bars (black) are the uncertainties associated with the irradiance pair method. The green symbols show the in situ extinction weighted average SSA throughout the spiral profile with standard deviations shown as error bars. c) The retrieved asymmetry parameter with associated error bars for both cases. d) The AAOD spectra from which the absorbing Ångström exponent is derived for both cases.**



a)

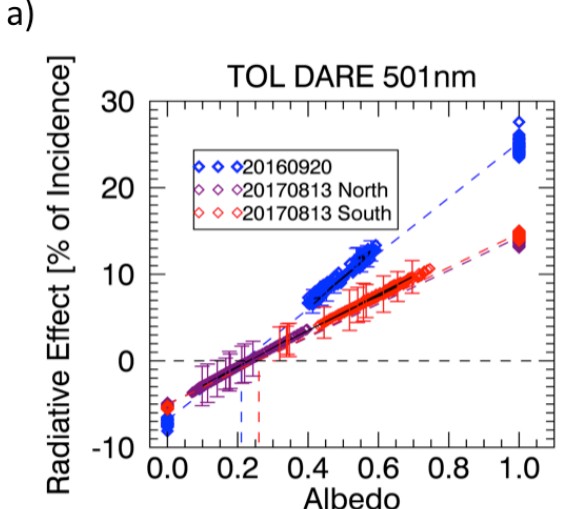

b)

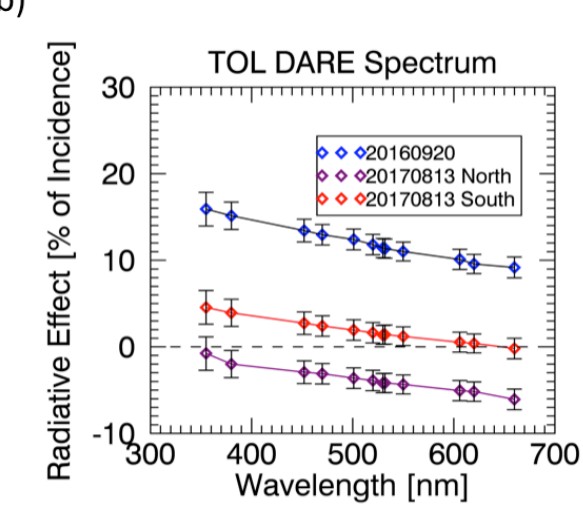

**Figure 8. a) The top of layer DARE at 501 nm as a function of the underlying albedo. The critical albedo at 501 nm is 0.2 across all three cases: 20 September 2016 in blue; 13 August 2017 North in purple; 13 August 2017 South in red. The uncertainty estimates are shown for a subset of data points for each case. b) An example of a DARE spectrum with associated uncertainties for all three cases: 20 September 20160 in blue; 13 August 2017 North in purple; 13 August 2017 South in red. The error bars slightly decrease with increasing wavelength for each case.**





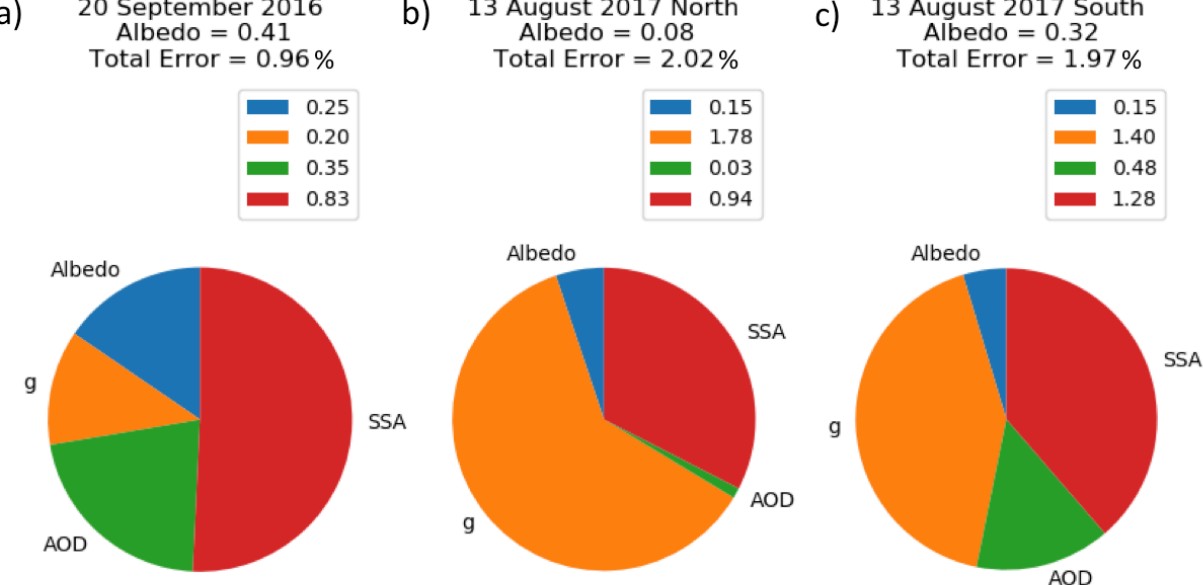

**Figure 9: The error contributions of g (orange), AOD (green), albedo (blue), and SSA (red) for one example each from a) 20 September 2016, b) 13 August 2017 North and c) 13 August 2017 South. Units are percentage of incident radiation.**



| Date | September 20th, 2016 | August 13th, 2017 |
|---|---|---|
| UTC | [11:55,12:14] | [10:05,10:15] |
| Latitude Range | [-16.79, -16.61] | [-9.02, -8.90] |
| Longitude Range | [8.80,8.99] | [4.88,5.00] |
| Albedo [501 nm] | 0.45 | 0.70 |
| Solar Zenith Angle | 21.0 | 33.5 |
| | | |

**Table 1: Case Description: Spiral**

| Date | September 20th, 2016 | August 13th, 2017 North | August 13th, 2017 South |
|---|---|---|---|
| UTC | [11:36, 11:42] | [12:05, 12:20] | [11:42,11:54] |
| Latitude Range | [-17.12,-16.97] | [-7.37, -6.29] | [-8.92, -8.06] |
| Longitude Range | [8.99,9.00] | [4.31,4.53] | [4.69,4.88] |
| Albedo Range [501nm] | [0.39,0.59] | [0.06, 0.39] | [0.29, 0.75] |
| Solar Zenith Angle Range | [18.5,18.8] | [22.1, 22.3] | [22.7, 23.5] |

**Table 2: Case Description: BOL Leg of Radiation Wall**

| 20160920 | Spiral | Wall (minimum, maximum) |
|---|---|---|
| $H_\infty$ | 0.0112 | -0.15, 0.11 |
| $i_\lambda$ 355 nm | 0.04 | -0.45,0.46 |
| $i_\lambda$ 532 nm | 0.08 | -0.86, 0.78 |
| $i_\lambda$ 1650 nm | 0.55 | -113.9, 100.2 |

5    **Table 3a. $H_\infty$ and select $i_\lambda$ values for 20 September 2016 case.**

| 20170813 | Spiral | Wall (minimum, maximum) |
|---|---|---|
| $H_\infty$ | 0.0131 | south:-0.65, 0.06<br>north:-0.83, -0.29 |
| $i_\lambda$ 355 nm | 0.08 | South: -1.9,0.35<br>North:-2.22,-1.35 |
| $i_\lambda$ 532 nm | 0.17 | South: -5.4,0.59<br>North: -5.43,-2.68 |
| $i_\lambda$ 1650 nm | 1.57 | South: -726.4, 3832.95<br>North: -410.8, -4.94 |

**Table 3b. $H_\infty$ and select $i_\lambda$ values for 13 August 2017 case.**



| $i_\lambda$ | ±1 | < 1 <br> > -1 | >1 <br> <-1 |
|---|---|---|---|
| Relative Magnitude | $V_\lambda \sim H_\lambda$ | $V_\lambda > H_\lambda$ | $V_\lambda < H_\lambda$ |
| Successful aerosol retrieval | Unlikely | Likely | Not possible |

**Table 4** Interpretation of $i_\lambda$ relating to the relative magnitudes of $V_\lambda, H_\lambda$.

| | SSFR-20160920 | SSFR-20170813 | In Situ-20160920 | In Situ-20170813 | Russell et al., 2002-SAFARI 2000 campaign |
|---|---|---|---|---|---|
| SSA-500 nm | 0.85 ±0.01 | 0.82±0.05 | | | 0.87 |
| SSA-530 nm | 0.84±0.01 | 0.81±0.05 | 0.86 ±0.03 | 0.88±0.01 | |
| AAE | 1.29 (355-660nm) | 1.44 (355-660nm) | 1.79 ±0.15 (470-660nm) | 1.71 ±0.07 (470-660 nm) | 1.45 (325-1000 nm) |

**Table 5. Comparison of ORACLES SSA and AAE values to Russell et al., 2010 SAFARI results. SSFR results include their estimated uncertainties; the in situ extinction weighted averages include corresponding standard deviations.**

## 5  Appendix A: Uncertainty Estimates

This appendix describes the full methodology used to obtain the uncertainties presented in the main body of this work. Some equations are repeated from the main body, and are included in an effort to make the derivation comprehensible.

The uncertainty analysis provides a way for us to evaluate our absorptance derivation methods, SSA
retrieval, and to assess if our DARE calculations are, in fact, more successful than existing methods.
Though we do not use radiation wall irradiance pairs to find aerosol intensive properties due to the
inability to separate *V* from *H*, we perform the uncertainty analysis for illustration only where we must
inaccurately assume *H=0.* We derive the uncertainty for both the radiation wall and spiral methods of
finding absorption and propagate those errors into the error of SSA. We assume measurements with
independent and random uncertainties, and therefore propagate errors by adding in quadrature.



## A.1 Absorptance

The uncertainty on absorptance is calculated from two separate methods: the irradiance pair method, completed using measurements from the radiation wall, and the spiral method which uses data taken only from the aircraft spiral.

The irradiance pairs method relies on determining the vertical flux divergence ($V_\lambda$) from two collocated irradiance measurement pairs above and below the aerosol layer ($F^\downarrow_{\lambda,top}$, $F^\uparrow_{\lambda,top}$, and $F^\downarrow_{\lambda,bot}$, $F^\uparrow_{\lambda,bot}$) following:

$$V_\lambda = \frac{(F^{net}_{\lambda,top} - F^{net}_{\lambda,bot})}{F^\downarrow_{\lambda,top}} = \frac{\left(F^\downarrow_{\lambda,top} - F^\uparrow_{\lambda,top}\right) - (F^\downarrow_{\lambda,bot} - F^\uparrow_{\lambda,bot})}{F^\downarrow_{\lambda,top}}. \tag{A1}$$

The absorptance would in theory (Song et al., 2016) be found by subtracting the horizontal photon
transport from $V_\lambda$:

$$A_\lambda = V_\lambda - H_\lambda. \tag{A2}$$

We assume that $H_\lambda = 0$ for the purposes of deriving a nominal uncertainty value, though this is an inappropriate assumption for the conditions encountered during ORACLES. We have no way of correcting for $H_\lambda$ and therefore the following calculations represent the nominal case where cloud
variability has no effect. With this assumption, the absorptance becomes:

$$A_\lambda = \frac{\left(F^\downarrow_{\lambda,top} - F^\uparrow_{\lambda,top}\right) - (F^\downarrow_{\lambda,bot} - F^\uparrow_{\lambda,bot})}{F^\downarrow_{\lambda,top}}, \tag{A3}$$

and the uncertainty is calculated as:

$$\delta A_\lambda = \sqrt{\left(\frac{-1}{F^\downarrow_{\lambda,top}} \delta F^\uparrow_{\lambda,top}\right)^2 + \left(\frac{-1}{F^\downarrow_{\lambda,top}} \delta F^\downarrow_{\lambda,bot}\right)^2 + \left(\frac{-1}{F^\downarrow_{\lambda,top}} \delta F^\uparrow_{\lambda,bot}\right)^2 + \left(\frac{-1}{F^\downarrow_{\lambda,top}} \delta F^\downarrow_{\lambda,top}\right)^2}, \tag{A4}$$

where $\delta F$ is the upper limit of the SSFR radiometric uncertainty, 5%. The uncertainty depends on the
magnitude of the downwelling irradiance, which is demonstrated clearly in Figure A1 for which the shorter wavelengths, where the incoming spectrum is the largest, have much larger uncertainties than the longer wavelengths.



The spiral method is based on many measurements throughout the profile of the atmospheric column, and we therefore rely on linearly fitting weighted AOD and irradiance measurements to determine the top of aerosol layer and bottom of aerosol layer net irradiances.

The following linear fits determine the irradiance values (upwelling/downwelling) at the top ($AOD_{532}$=minimum) and bottom of the aerosol layer ($AOD_{532}$= maximum):

$$F_\lambda^\uparrow = a_\lambda^\uparrow + b_\lambda^\uparrow * AOD_{532}, \tag{5A}$$

$$F_\lambda^\downarrow = a_\lambda^\downarrow + b_\lambda^\downarrow * AOD_{532}, \tag{6A}$$

where $a_\lambda$ and $b_\lambda$ are the slope and intercept of the linear fit lines.

The uncertainties on the weighted fit parameters $a_\lambda$, $b_\lambda$ are calculated according to:

$$\sigma_a = \frac{\sum w*(AOD_{532})^2}{\Delta}, \tag{7A}$$

$$\sigma_b = \frac{\sum w}{\Delta}, \tag{8A}$$

$$w = \frac{1}{\sigma_i^2}, \tag{9A}$$

where $\Delta = \sum w * \sum w * (AOD_{532})^2 - (\sum w * AOD_{532})^2$ and $\sigma_i$ represents the measurement error. Therefore, when substituting these into equation 3, the absorptance can be found by:

$$A_\lambda = \frac{AOD_{532}^{max}*(b_\lambda^\uparrow - b_\lambda^\downarrow)}{a_\lambda^\downarrow}, \tag{10A}$$

and the uncertainty on the absorptance is:

$$\delta A_\lambda = \sqrt{\left(\frac{dA_\lambda}{dAOD_{532}^{max}} * \delta AOD_{532}^{max}\right)^2 + \left(\frac{dA_\lambda}{da_\lambda^\downarrow} * \sigma_{a,\lambda}^\downarrow\right)^2 + \left(\frac{dA_\lambda}{db_\lambda^\uparrow} * \sigma_{b,\lambda}^\uparrow\right)^2 + \left(\frac{dA_\lambda}{db_\lambda^\downarrow} * \sigma_{b,\lambda}^\downarrow\right)^2}, \tag{11A}$$

where $\sigma_{a,b}^{\downarrow,\uparrow}$ are the uncertainties on the linear fit parameters. The spiral method compared to the radiation wall method reduces the absorptance uncertainty from 0.05 to 0.02 at 501 nm for 20 September 2016, which is visualized in Figure A1, and from 0.07 to 0.05 at 501 nm for 13 August 2017.

## A.2 SSA

The SSA uncertainty from the spiral measurements is a product of the retrieval while the absorptance error is propagated into the SSA calculation for the wall illustration. The comparison is shown in





Figures 8a and 8b where it is clear to see that the spiral method significantly decreases the SSA uncertainty estimates compared to the irradiance pairs method. The uncertainty from the irradiance pairs method would be even larger if we considered the uncertainty due to non-zero H.

The radiation wall SSA uncertainty is calculated according to:

$$\delta SSA_\lambda = \sqrt{\left(\frac{dSSA_\lambda}{dAOD_\lambda} * \delta AOD_\lambda\right)^2 + \left(\frac{dSSA_\lambda}{dA_\lambda} * \delta A_\lambda\right)^2} ,$$          (12A)

where $\frac{dSSA_\lambda}{dAOD_\lambda} = \left(\frac{-1}{AOD_\lambda^2}\right) * \frac{\mu * \ln\left(1 - \frac{A_\lambda}{c_{1,\lambda}}\right)}{c_{2,\lambda}}$ and $\frac{dSSA}{dA_\lambda} = \left(\frac{\mu}{AOD_\lambda * c_{2,\lambda}}\right) * \left(\frac{1}{A_\lambda - c_{1,\lambda}}\right)$.

To simplify propagation of errors, we determine the relationship between absorptance and AAOD by an exponential fit determined through 1D radiative transfer calculations:

$$A_\lambda = c_1 \left(1 - e^{-c_2 * \frac{AAOD_\lambda}{\mu}}\right),$$          (13A)

where $\mu = \cos(sza)$.

The constants c1 and c2 are presented in appendix table 1A and table 2A, and an example of the exponential fit at 380 nm between absorptance and AAOD is shown in appendix figure A2.

The Absorption Angstrom Exponent (AAE) and aerosol absorption optical depth (AAOD) are determined as follows:

*AAOD=(1-SSA)\*AOD,*          (14Aa)

$$AAOD = \left(AAOD_{500} * \frac{\lambda}{\lambda_{500}}\right)^{-AAE}.$$          (14Ab)

Equation 14 can be combined with 15a and solved for SSA:

$$SSA_\lambda = 1 + \frac{\mu * \ln\left(1 - \frac{A_\lambda}{c_1}\right)}{AOD_\lambda * c_2},$$          (15A)

Equation 16A provides us with a relationship for which we can calculate the uncertainty on SSA

(equation 13A).




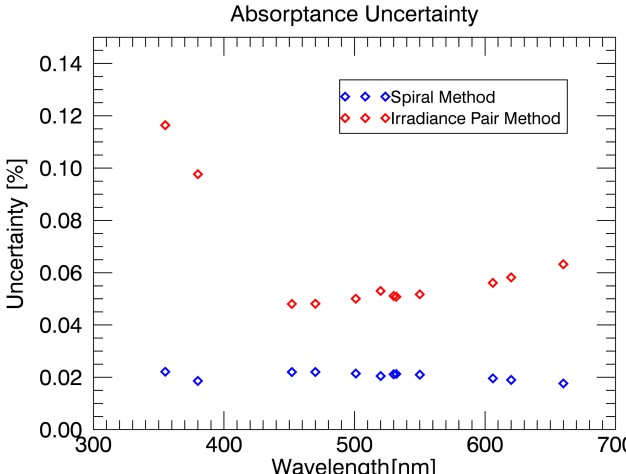

**Figure A1:** The uncertainty values for the absorptance derivation from the spiral method, shown in blue, and the irradiance pairs method, shown in red, for 20 Septmber 2016. The figure is similar for 13 August 2017. The uncertainty is significantly reduced with the spiral method, especially at the shortest wavelengths where the incoming irradiance is largest. The uncertainty estimate for the irradiance pairs method depends upon the value of the incoming irradiance (equation 7), which is largest at the shortest wavelengths.

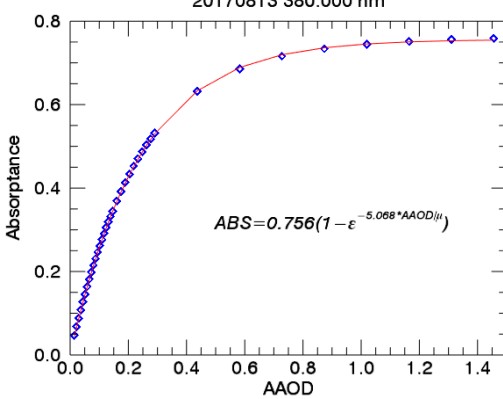

**Figure A2:** Radiative transfer calculations at 380 nm of the relationship between absorptance and AAOD. The constant values c1 and c2 for this case at this wavelength are 0.756 and -5.086, respectively.

| Wavelength | C1 | C2 |
|---|---|---|





| [nm] | | |
|------|-------|--------|
| 355 | 0.797 | -2.811 |
| 380 | 0.794 | -2.836 |
| 452 | 0.794 | -2.794 |
| 470 | 0.797 | -2.773 |
| 501 | 0.798 | -2.763 |
| 520 | 0.797 | -2.77 |
| 530 | 0.797 | -2.767 |
| 532 | 0.796 | -2.773 |
| 550 | 0.80 | -2.75 |
| 606 | 0.805 | -2.711 |
| 620 | 0.803 | -2.727 |
| 660 | 0.812 | -2.666 |

**Appendix Table 1: Constant values c1 and c2 determined by radiative transfer calculations for equation 15A for 20 September 2016.**

| Wavelength [nm] | C1 | C2 |
|-----------------|-------|-------|
| 355 | 0.761 | 5.147 |
| 380 | 0.756 | 5.068 |
| 452 | 0.777 | 4.812 |
| 470 | 0.785 | 4.759 |
| 501 | 0.788 | 4.739 |
| 520 | 0.793 | 4.710 |
| 530 | 0.794 | 4.709 |
| 532 | 0.793 | 4.705 |
| 550 | 0.801 | 4.636 |
| 606 | 0.819 | 4.465 |
| 620 | 0.819 | 4.479 |
| 660 | 0.835 | 4.321 |

5   **Appendix Table 2: Constant values c1 and c2 determined by radiative transfer calculations for equation 15A for 13 August 2017.**