# Peer review of "Above-Cloud Aerosol Radiative Effects based on ORACLES 2016 and ORACLES 2017 Aircraft Experiments"

_Atmospheric Measurement Techniques, 2019_

## Referee Comment (RC1) · Anonymous Referee #1 · 12 Jul 2019

General comments

The authors focus on the calculation of the direct aerosol radiative effect over bright clouds using airplane campaign measurements. They propose a new methodology to derive the vertically resolved aerosol properties, minimizing horizontal cloud inhomogeneity. They identify a critical cloud albedo value and compare their findings with past studies. I enjoyed reading their work, because the subject is scientifically important, the presentation very clear and the scientific methods seem robust. It is a very good manuscript, which I find publishable with only minor changes. They are only secondary scientific points, whose resolution will not alter the findings of the study. Moreover, there

are only a few technical corrections.

Specific comments

p. 2, ll. 10-11. I would write this as "... radiative effect occurs at an albedo value (critical albedo) just above 0.2 ..."

p. 3, l. 22. A couple of more recent works that might be inserted are: Oikawa, E., Nakajima, T., Winker, D., 2018. An evaluation of the shortwave direct aerosol radiative forcing using CALIOP and MODIS observations. J. Geophys. Res. Atmos. 123 (2), 1211–1233. Korras-Carraca, M. B., Pappas, V., Hatzianastassiou, N., Matsoukas, C., 2019. Global vertically resolved aerosol direct radiation effect from three years of CALIOP data using the FORTH radiation transfer model, Atmospher. Res., 224, 138-156

p. 4, l. 4. The reader gets the erroneous impression the the Kim et al. correction to CALIOP is unofficial. I would use the phrase "... until the development of a new method in version 4 to derive AOD ..."

p. 8, l. 7. "The nadir light collector is not actively leveled". Just for clarity reasons, please state if the upwelling flux is sensitive or not to the pitch and roll angles.

p. 10, ll. 21-24. Why not use a circular pattern with smaller, within the ALP limits, pitch and roll angles? Would the area covered be too large then?

p. 11, ll. 3-5. Surely the spatial variability is smaller with the spiral descent. However, only one albedo value is reported in Table 1. Shouldn't there be a range of albedos from all the upwelling-downwelling pairs?

Figure 2. I assume that all points in 2a,c are altitude-filtered, since the altitude filter is not mentioned in the color scheme. In that case, the caption of a) and c) could be "The latitude vs. altitude of altitude-filtered ..."

Figure 2. Where are in a) the purple dots between latitudes -16.65 and -16.70 as seen

in b)? Similarly, where are in c) the -8.9 latitude purple dots shown in d)? I cannot detect the correspondence between the points of a and b and between c and d. It would be better if in all figures, the start and end of the spiral were marked clearly.

Figure 4a. Unless I missed it in the text, the 470, 530, 660 nm data points are never explained. They probably belong to the 2016 case, but I am not sure. Also, which wavelength corresponds to the blue and red points?

p. 15, l. 5. If I understand it correctly, for the 2016 case H_infinity is less than 1.12 %, not less than 0.2 %. Please clarify.

p. 15, l. 20. $H\lambda$ is not defined rigorously, so we are not sure if $A\lambda=V\lambda+H\lambda$ or $A\lambda=V\lambda-H\lambda$. It is mentioned in the Appendix, however.

p. 15, ll. 25-26. Because of the non-rigorous definition of $H\lambda$, we just have to trust the authors here.

p. 17, l. 18. In the beginning I was confused by how different the profiles of AOD and extinction coefficient were in Figure 5. I then realized that of course AOD is the 4STAR column-integrated AOD down to that height, while the extinction is local. So I suggest that this line be changed to "... so that the column-integrated AOD profile decreases ...", just to remind the reader.

p. 17, l. 23. If I understand correctly, the extinction coefficient is derived from the 4STAR AOD data. Is it meaningful to compare the extinction coefficient with measure-ments from the HSRL-2 instrument? Such measurements exist for the 2016 case, don't they? Under the same light, where have the HSRL-2 data been used? Could the HRSL-2 be removed from the description altogether?

p. 18, l. 23. "... retrieval at 501 nm ...". In Fig. 6b the title is "380 nm".

p. 24, l. 12. Here as also in l. 8 of the previous page, the albedos given do not match the albedos of Table 2. Are we referring to the TOL sweep?

[Figure]

p. 48., l. 6. These derivatives come from Equations 13A and 8A, so it would be clearer if 13A were presented first.

Technical corrections

p. 22, l. 2. "... in Figure 8..." I think the authors mean Figure 7.

Figure A1. Equation 7 is mentioned, but it is irrelevant

p. 48, l. 1. "Figures 8a and 8b ..." probably should be Figures 7a and 7b

p. 48, l. 19. There is no Equation 16A. Generally, the A2 part of the Appendix should be reviewed and polished.

---

## Referee Comment (RC2) · Anonymous Referee #2 · 18 Jul 2019

**General comments**

This paper introduces an approach to derive aerosol properties from aircraft measurements in the presence of underlying clouds, which normally bias the retrieval. Although I'm not an expert on aircraft measurements, the manuscript is easy to read. The sections can be better structured though, separating methods and results better. The manuscript is well suited for publication in AMT, and I recommend publication after a few minor modifications, which focus mainly on the clarification of the text, to make it more accessible for the general audience. It is up to the authors to follow these recommendations.

[Figure]

Recommendations:

Section 1.2. There are two references in the publication list that are not mentioned in the text: Peers, 2015 and De Graaf, 2012. These two publications describe DARE retrievals from specific instruments which are unique and useful in their own ways for DARE retrievals from satellite platforms (polarization measurements from POLDER, and hyperspectral measurements from SCIAMACHY). It makes sense to briefly mention these studies in this section.

It would be useful to have an idea of the situation on the two selected days during which the measurements were obtained. Section 2.5 and figure 2 give the technical details and the exact locations, but it would be nice to have a general idea of the situation: Are we looking at smoke above clouds, which I know was the general goal for the ORACLES campaigns? Where was the smoke coming from, was it aged, how far are we from the coast? How closed was the cloud field, which is rather relevant in this study? A figure showing e.g. satellite overview and flight track may help, and a general description of the meteorological situation would be nice.

I belief p13, l.22 – p14, l.16 should be part of section 3 and described before section 3.1.

I found section 3.1.1. hard to understand. The relevance of $H_\lambda$ is clear, but the determination of $H_\infty$ is not clear to me. I don't understand the legend and the symbols used in Figure 4a. How and why are the dates and the wavelengths mixed?

Section 3.2: p18, l23: Figure 6b shows the example at 380nm according to the title of the figure.

Section 4.1: The authors describe figure 7 here, but switch to Figure 8 on p22, l.2. This should probably be Figure 7 as well.

App. A.2 The SSA is described here, depicted in Figure 7, not 8. Do the authors mean Figure 7a and 7b on p. 48?

---

## Author Comment (AC1) · 15 Aug 2019

We would like to thank both reviewers for their valuable and constructive comments. The manuscript has been edited in response to these comments to the extent we believe improves the scientific understanding. Please find the reviewer's comments and our responses below. A revised manuscript is attached, followed immediately by the edited manuscript showing all changes. Unless specifically stated, the line numbers cited in our responses refer to the revised manuscript.

Anonymous Referee #1 General comments

[Figure]

The authors focus on the calculation of the direct aerosol radiative effect over bright clouds using airplane campaign measurements. They propose a new methodology to derive the vertically resolved aerosol properties, minimizing horizontal cloud inhomogeneity. They identify a critical cloud albedo value and compare their findings with past studies. I enjoyed reading their work, because the subject is scientifically important, the presentation very clear and the scientific methods seem robust. It is a very good manuscript, which I find publishable with only minor changes. They are only secondary scientific points, whose resolution will not alter the findings of the study. Moreover, there are only a few technical corrections.

Thank you for your positive view of our manuscript.

Specific comments p. 2, ll. 10-11. I would write this as "... radiative effect occurs at an albedo value (critical albedo) just above 0.2 ..." Thank you for this suggestion, we have implemented these changes on p.2, l. 14

p. 3, l. 22. A couple of more recent works that might be inserted are: Oikawa, E., Nakajima, T., Winker, D., 2018. An evaluation of the shortwave direct aerosol radiative forcing using CALIOP and MODIS observations. J. Geophys. Res. Atmos. 123 (2), 1211–1233. Korras-Carraca, M. B., Pappas, V., Hatzianastassiou, N., Matsoukas, C., 2019. Global vertically resolved aerosol direct radiation effect from three years of CALIOP data using the FORTH radiation transfer model, Atmospher. Res., 224, 138-156 Thank you, we have included these works on p. 3 l. 26/27

p. 4, l. 4. The reader gets the erroneous impression the the Kim et al. correction to CALIOP is unofficial. I would use the phrase "... until the development of a new method in version 4 to derive AOD ..." Thank you, this has been added on p. 4 1. 8/9

p. 8, l. 7. "The nadir light collector is not actively leveled". Just for clarity reasons, please state if the upwelling flux is sensitive or not to the pitch and roll angles. We have added information on the nadir light collector for clarity on p. 8 l. 25-27

p. 10, ll. 21-24. Why not use a circular pattern with smaller, within the ALP limits, pitch and roll angles? Would the area covered be too large then?

The pitch and roll limits for the leveling platform are about 6 deg. Typically, spirals are flown with bank (roll) angles of ∼30deg, which is much larger than the limits of the platform. Indeed, a circle with a bank angle under the roll limit would be too large. We have included additional information on the use of the ALP on p. 11, l. 23-27

p. 11, ll. 3-5. Surely the spatial variability is smaller with the spiral descent. However, only one albedo value is reported in Table 1. Shouldn't there be a range of albedos from all the upwelling-downwelling pairs? The albedo value reported in Table 1 is the albedo measured just above the cloud top. While we could report the range of albedos measured throughout the spiral, we choose to report the cloud top albedo only since that is the albedo value used in the aerosol retrieval. We have made this clearer within the text on p. 12, l. 12-23 and in Table 1.

Figure 2. I assume that all points in 2a,c are altitude-filtered, since the altitude filter is not mentioned in the color scheme. In that case, the caption of a) and c) could be "The latitude vs. altitude of altitude-filtered ..." Thank you for pointing out the ambiguity of the figure caption. We have added the descriptor in the caption of Figure 2.

Figure 2. Where are in a) the purple dots between latitudes -16.65 and -16.70 as seen in b)? Similarly, where are in c) the -8.9 latitude purple dots shown in d)? I cannot detect the correspondence between the points of a and b and between c and d. It would be better if in all figures, the start and end of the spiral were marked clearly. Thank you for the feedback on this figure. We have updated the figure and included longitude vs. altitude along with latitude vs. altitude for more clarity on the corresponding points between plots.

Figure 4a. Unless I missed it in the text, the 470, 530, 660 nm data points are never explained. They probably belong to the 2016 case, but I am not sure. Also, which wavelength corresponds to the blue and red points? Thank you for pointing this out. We

[Figure]

have updated Figure 4 along with the caption for more clarity. 20160920 is presented in blue; 20170813 in red; a wall example from 20160920 is presented in green. Each case uses all of the 4STAR wavelengths to derive H_infinity. The specific wavelengths of 452 and 865 nm are highlighted to illustrate that the full spectrum is used.

p. 15, l. 5. If I understand it correctly, for the 2016 case H_infinity is less than 1.12 %, not less than 0.2 %. Please clarify. Thank you for pointing this out. The H_inifinity values are 0.0112 (1.12%) and 0.0131(1.31%) for 20160920, 20170813 cases respectively. The text has been updated on p. 16, l. 8/9

p. 15, l. 20. H$\lambda$ is not defined rigorously, so we are not sure if A$\lambda$=V$\lambda$+H$\lambda$ or A$\lambda$=V$\lambda$-H$\lambda$. It is mentioned in the Appendix, however. We have included the specific definition of A$_\lambda$= V$_\lambda$-H$_\lambda$ on p.11, l 3. This is the definition as used in Schmidt et al., 2010 and Song et al., 2016. This deviates from other definitions in the literature, but we wanted to stay consistent with previous papers.

p. 15, ll. 25-26. Because of the non-rigorous definition of H$\lambda$, we just have to trust the authors here. The definition has been added on p.11 l. 3

p. 17, l. 18. In the beginning I was confused by how different the profiles of AOD and extinction coefficient were in Figure 5. I then realized that of course AOD is the 4STAR column-integrated AOD down to that height, while the extinction is local. So I suggest that this line be changed to "... so that the column-integrated AOD profile decreases ...", just to remind the reader. Thank you for this suggestion. The text now reflects this addition on p.18 l. 20

p. 17, l. 23. If I understand correctly, the extinction coefficient is derived from the 4STAR AOD data. Is it meaningful to compare the extinction coefficient with measurements from the HSRL-2 instrument? Such measurements exist for the 2016 case, don't they? Under the same light, where have the HSRL-2 data been used? Could the HRSL-2 be removed from the description altogether? The 4STAR AOD data is used to calculate extinction only for the spiral measurements to be used within the aerosol

retrieval. The downward-viewing HSRL-2 data is used to calculate DARE from the wall measurements for which we do not have 4STAR AOD throughout the column to derive extinction. We have updated the methods on p. 19 l.# to state this more explicitly. While it may be a meaningful exercise to compare 4STAR to HSRL-2 data, we leave that specific analysis out of manuscript but have included a note on p. 19, l. 3-5

p. 18, l. 23. "... retrieval at 501 nm ...". In Fig. 6b the title is "380 nm". Thank you for noting this. The text has been updated to 380 nm on p. 19 l. 27

p. 24, l. 12. Here as also in l. 8 of the previous page, the albedos given do not match the albedos of Table 2. Are we referring to the TOL sweep? The albedo values are what we consider the 'scene albedo', or the average of the entire range measured across the BOL leg of the radiation wall. Thank you for pointing out the slight differences between the albedos presented in the text vs. table 2; we have updated the text to reflect the correct albedo values. We have also included a reminder to the reader that the albedo measurements are taken from the BOL leg of the radiation wall on p. 25 l. 19

p. 48., l. 6. These derivatives come from Equations 13A and 8A, so it would be clearer if 13A were presented first. The order of equations presented in A2 part of the Appendix has been updated. P. 49

Technical corrections

p. 22, l. 2. "... in Figure 8..." I think the authors mean Figure 7. Thank you for catching this, text is revised on p. 23 1. 11

Figure A1. Equation 7 is mentioned, but it is irrelevant The mention of Equation 7 has been removed in the caption of Figure A1.

p. 48, l. 1. "Figures 8a and 8b ..." probably should be Figures 7a and 7b Thank you for pointing this out, figure numbers have been updated on p. 51 1. 21/22

p. 48, l. 19. There is no Equation 16A. Generally, the A2 part of the Appendix should be reviewed and polished. The equation numbers have been edited to reflect the

correct equations. The Appendix has been reviewed, with changes made to increase the clarity on p. 48-51

Please also note the supplement to this comment:
https://www.atmos-meas-tech-discuss.net/amt-2019-125/amt-2019-125-AC1-supplement.pdf

———————————————————

[Figure]

**Supplement:**

[revised manuscript text omitted]

---

## Author Comment (AC2) · 15 Aug 2019

We would like to thank both reviewers for their valuable and constructive comments. The manuscript has been edited in response to these comments to the extent we believe improves the scientific understanding. Please find the reviewer's comments and our responses below. A revised manuscript is attached, showing all changes. Unless specifically stated, the line numbers cited in our responses refer to the revised manuscript.

Anonymous Referee #2

[Figure]

General comments This paper introduces an approach to derive aerosol properties from aircraft measurements in the presence of underlying clouds, which normally bias the retrieval. Although I'm not an expert on aircraft measurements, the manuscript is easy to read. The sections can be better structured though, separating methods and results better. The manuscript is well suited for publication in AMT, and I recommend publication after a few minor modifications, which focus mainly on the clarification of the text, to make it more accessible for the general audience. It is up to the authors to follow these recommendations.

Thank you for the recommendations. We have made slight modifications to the paper, such as including additional subheadings and increasing contextual information, to increase clarity. However, we choose to keep the separation of methods and results as is.

Recommendations:

Section 1.2. There are two references in the publication list that are not mentioned in the text: Peers, 2015 and De Graaf, 2012. These two publications describe DARE retrievals from specific instruments which are unique and useful in their own ways for DARE retrievals from satellite platforms (polarization measurements from POLDER, and hyperspectral measurements from SCIAMACHY). It makes sense to briefly mention these studies in this section. Thank you for this suggestion. We have included a brief description of these studies, as well as a new paper [de Graaf et al., 2019] that just came out in ACP on p. 5, l. 3-14

It would be useful to have an idea of the situation on the two selected days during which the measurements were obtained. Section 2.5 and figure 2 give the technical details and the exact locations, but it would be nice to have a general idea of the situation: Are we looking at smoke above clouds, which I know was the general goal for the ORA- CLES campaigns? Where was the smoke coming from, was it aged, how far are we

from the coast? How closed was the cloud field, which is rather relevant in this study? A figure showing e.g. satellite overview and flight track may help, and a general description of the meteorological situation would be nice. Thank you for this suggestion. While we agree that it would be beneficial to include the suggested information, many of the elements are currently being studied in other analyses from the oracles campaign and outside the scope of this paper (i.e. aerosol age, meteorological conditions, aerosol back-trajectories). We have included more situational information on both the clouds and aerosols in section 2.5 on p 12. We have chosen not to include a satellite overview figure, since the conditions change so rapidly we have concerns that it would give the reader an inaccurate description of the conditions. Figures 2b and 2d include cloud imagery from eMAS (b) and SEVIRI (c) with the spiral flight track overlaid, which give the reader a sense of the cloud field during the spiral.

I belief p13, l.22 – p14, l.16 should be part of section 3 and described before section 3.1.

We understand that this refers to "To ensure that the aerosol signal is isolated. . ." and the following text from 3.1. We considered moving this section to before section 3.1 (for example, to section 2), but then found that it makes the most sense to talk about the filtering and separation of cloud/aerosol signal after introducing the walls and spiral sampling strategy. We added a sub-heading to make this more clear (3.1.1: Data filtering; 3.1.2: Horizontal Flux Divergence, as before).

I found section 3.1.1. hard to understand. The relevance of $H\lambda$ is clear, but the determination of $H\infty$ is not clear to me. I don't understand the legend and the symbols used in Figure 4a. How and why are the dates and the wavelengths mixed? We have updated Figure 4 along with the caption for more clarity. 20160920 is presented in blue; 20170813 in red; a wall example from 20160920 is presented in green. Each case uses all of the 4STAR wavelengths to derive H_infinity. The specific wavelengths of 452 and 865 nm are highlighted to illustrate that the full spectrum is used.

[Figure]

Section 3.2: p18, l23: Figure 6b shows the example at 380nm according to the title of the figure. Thank you for pointing this out. The text has been updated on p19 l. 27

Section 4.1: The authors describe figure 7 here, but switch to Figure 8 on p22, l.2. This should probably be Figure 7 as well. The figure number has been updated in the text on p. 23, l. 11

App. A.2 The SSA is described here, depicted in Figure 7, not 8. Do the authors mean Figure 7a and 7b on p. 48? Yes, it should be Figure 7, thank you for pointing out the inconsistency. App. A.2 has been updated significantly for clarity and correctness on p. 48-51.

Please also note the supplement to this comment:
https://www.atmos-meas-tech-discuss.net/amt-2019-125/amt-2019-125-AC2-supplement.pdf

⎯⎯⎯⎯⎯⎯⎯⎯⎯⎯⎯⎯

**Supplement:**

[revised manuscript text omitted]